

# Calibrating a global three-dimensional biogeochemical ocean model (MOPS-1.0)

Iris Kriest[1], Volkmar Sauerland[2], Samar Khatiwala[3], Anand Srivastav[2], and Andreas Oschlies[1]

[1]GEOMAR Helmholtz-Zentrum für Ozeanforschung Kiel, Düsternbrooker Weg 20, D-24105 Kiel, Germany
[2]Institut für Informatik, Christian-Albrechts-Universität zu Kiel, Christian-Albrechts-Platz 4, D-24098 Kiel, Germany
[3]Department of Earth Sciences, University of Oxford, South Parks Road, Oxford OX1 3AN, UK

*Correspondence to:* Iris Kriest (ikriest@geomar.de)

**Abstract.**

Global biogeochemical ocean models contain a variety of different biogeochemical components and often much simplified representations of complex dynamical interactions, which are described by many ($\approx 10- \approx 100$) parameters. The values of many of these parameters are empirically difficult to constrain, due to the fact that in the models they represent processes for a range of different groups of organisms at the same time, while even for single species parameter values are often difficult to determine in situ. Therefore, these models are subject to a high level of parametric uncertainty. This may be of consequence for their skill with respect to accurately describing the relevant features of the present ocean, as well as their sensitivity to possible environmental changes.

We here present a framework for the calibration of global biogeochemical ocean models on short and long time scales. The framework combines an offline approach for transport of biogeochemical tracers with an Estimation of Distribution Algorithm (Covariance Matrix Adaption Evolution Strategy, CMAES). We explore the performance and capability of this framework by five different optimizations of six biogeochemical parameters of a global biogeochemical model. First, a twin experiment explores the feasibility of this approach. Four optimizations against a climatology of observations of annual mean dissolved nutrients and oxygen determine the extent, to which different setups of the optimization influence model's fit and parameter estimates. Because the misfit function applied focuses on the large-scale distribution of inorganic biogeochemical tracers, parameters that act on large spatial and temporal scales are determined earliest, and with the least spread. Parameters more closely tied to surface biology, which act on shorter time scales, are more difficult to determine. In particular the search for optimum zooplankton parameters can benefit from a sound knowledge of maximum and minimum parameter values, leading to a more efficient optimization. It is encouraging that, although the misfit function does not contain any direct information about biogeochemical turnover, the optimized models nevertheless provide a better fit to observed global biogeochemical fluxes.

## 1 Introduction

Global ocean models that simulate biogeochemical interactions are subject to many uncertainties, among them those related to initial conditions, forcing, and parameterizations of physical and biological processes, as well as the adequacy of the chosen model complexity with respect to the scientific problem under investigation. It is generally assumed that all these 'input'



factors affect the simulation results in ways that may be different for different models, but a thorough understanding of how uncertainties in input map onto model output (residuals, i.e., deviations from the true state) is still lacking. Quantitative estimates of the effect of model uncertainty on model residuals are generally obtained from individual sensitivity studies, model intercomparison or model ensemble studies, where the spread of model results is regarded as a measure of model uncertainty.

This procedure is, for example, followed in the assessment reports of the Intergovernmental Project of Climate Change (IPCC). The Ocean Carbon Model Intercomparison Project (OCMIP, Orr et al., 2001) applied a strict protocol regarding the description of biogeochemical processes to a suite of different ocean circulation models to show that the effect of uncertainties in the simulated circulation on biogeochemical tracer distributions and their residuals can be considerable (Najjar et al., 2007). However, the effect of uncertainties in the formulation of biogeochemical models on simulated global biogeochemical tracers and fluxes

can be of similar magnitude (Kriest et al., 2010) and is often difficult to disentangle from other sources of uncertainty (e.g., Cabre et al., 2015; Seferian et al., 2015). One reason for diverging results of global biogeochemical models can be related to the uncertainty with respect to biological constants and equations. In addition to often poorly constrained parameters, it is, so far, not even clear how complex a biogeochemical model should be (e.g. what state variables it should contain) in order to realistically reproduce observed global tracer distributions (Kriest et al., 2012). As a consequence, the diversity of biogeo-

chemical models ranges from simple, "nutrient-only" models to far more complex ones, comprising different elemental cycles and biological components.

Uncertainties in biogeochemical model setup partly arise from sparse observations, particularly in the open ocean and during winter season in the high latitudes (Kriest et al., 2010). Further, the combined effects of shallow and deep biogeochemistry and the rather sluggish ocean circulation introduce a variety of timescales, from minutes to millennia, hampering a complete and

thorough investigation of the combined effects of the different process parameterizations. Finally, even quite simple biogeochemical models are often characterized by non-linear interactions, complicating the a posteriori analysis of model results. By performing a relatively "coarse sweep" of the multidimensional model parameter space, Kriest et al. (2010, 2012) illustrated the impact of different model complexities and parameter sets on simulated tracers and their fit to observations. This first attempt to systematically explore the impacts of biogeochemical parameter uncertainty in global models may well have missed optimal

regions in parameter space, making it difficult to decide whether a model performs badly due to ill-chosen parameters, or due to an insufficient model structure. The establishment of an automatic optimization of global biogeochemical ocean models is aimed for in this current study that should enable a more thorough search for "best" parameters, and thus facilitate inter-model comparison.

An under-sampled ocean, together with a large variety of time and space scales and a high level of structural model com-

plexity, poses a challenge for optimization, and for a full, and dense enough, scan of the parameter space on a global scale. Therefore, optimization of marine biogeochemical models has mostly been carried out in a local, 0- or 1-dimensional setting (e.g., Fasham and Evans, 1995; Athias et al., 2000; Rückelt et al., 2010; Ward et al., 2010). The variability of biogeochemical processes has been addressed by simultaneous optimization at different sites (and physical forcings) in the North Atlantic by Schartau and Oschlies (2003a, b). Given the high computational demands, and the sparsity of biogeochemical data on a global

scale, attempts to address the indeterminacy of global simulations of ocean biogeochemistry via optimization have resorted





to rather simple biogeochemical systems (Kwon and Primeau, 2006, 2008) or to rather coarse physical model environments (Tjiputra et al., 2007). To constrain parameters related to dissolved organic matter production and decay on short and long time scales, Letscher et al. (2015) alternated between a simplified biogeochemical system and a more complex model, which is limited in terms of spin-up time. Recent attempts begin to combine complex, local models and a detailed three-dimensional
global environment for optimization (Hemmings et al., 2014). To our knowledge, however, the experiments presented here are the first one that, for a state-of-the-art global biogeochemical ocean model, carry out a parameter optimization that targets at parameters relevant for biogeochemical processes on both large and small scales in the full spatio-temporal domain.

In this paper we first test the global biogeochemical model optimization against synthetic data, derived from a previous model experiment with perturbed model parameters in so-called twin experiments. We then present four optimizations against
a global, synoptic data set of observed phosphate, nitrate, and oxygen.

## 2 Methods

### 2.1 Biogeochemical ocean model

#### 2.1.1 Circulation framework

For easy and generic coupling between different biogeochemical models and circulation fields, as well as fast and efficient
computation we use the "Transport Matrix Method" (TMM), developed by Samar Khatiwala (Khatiwala, 2007), and available via Github (https://github.com/samarkhatiwala/tmm). This efficient "offline" method for ocean passive tracer transport represents the advective and diffusive components of an ocean circulation model in form of transport matrices, that have been extracted prior to the biogeochemical simulations performed here from a physical global circulation model.

For optimization, we use the TMM with monthly mean transport matrices derived from a 2.8° global configuration of the
MIT ocean model with 15 levels in the vertical (Marshall et al., 1997). Using this rather coarse spatial grid, a time step length of 1/2 day for tracer transport and 1/16 day for biogeochemical interactions, each biogeochemical model setup with seven tracers (Kriest and Oschlies, 2015) has been simulated for 3000 years, after which most of the tracers approach steady state (see also Kriest and Oschlies, 2015).

#### 2.1.2 Biogeochemical model

The biogeochemical model employed as representative of current state-of-the-art models is the same as presented by Kriest and Oschlies (2015, hereafter called MOPS), and we only describe it briefly here. It consists of seven tracers, namely phosphate, nitrate, phytoplankton, zooplankton, detritus, dissolved organic matter (DOM) and oxygen. For conversion between the different elements we apply a constant global stoichiometry of $R_{-O2:P} =170$ mmol $O_2$:mmol P for the ratio between $O_2$:P, and 16 mmol N:mmol P for the N:P ratio of particular and dissolved organic matter. The stoichiometry of aerobic and anaerobic
remineralization is based on Paulmier et al. (2009). Remineralization of detritus and DOM is parameterized via a constant nominal remineralization rate, $r = 0.05$ [d$^{-1}$]. However, aerobic remineralization is restricted to regions with sufficient oxygen. If



oxygen declines, nitrate is used as electron acceptor, thereby mimicking denitrification. If both oxygen and nitrate are depleted, remineralization of organic matter is suppressed in the model. Both aerobic and anaerobic remineralization are parameterized as a saturation curve, using half-saturation constants to regulate the affinity of these processes to either oxidant, as well as the inhibition of denitrification through oxygen. Thus, the accomplished remineralization rate may differ from $r$, depending on

oxidant availability. Temperature dependent nitrogen fixation resupplies fixed nitrogen lost through denitrification via relaxation at the sea surface to the stoichiometric ratio of 16. Thus, while total phosphate inventory is conserved, oxygen and fixed nitrogen inventory may change during the course of the simulation, with the long-term, steady state inventory depending on physics and biogeochemistry (Kriest and Oschlies, 2015).

Sinking of detritus is simulated using a sinking speed increasing with depth $w = a\,z$ [d$^{-1}$]. For better comparison to observed

particle flux profiles (e.g., Martin et al., 1987), in the following we express the sinking speed via the parameter $b = r/a$ (see Kriest and Oschlies, 2008). The model also includes burial of organic matter arriving at the sea floor, which is resupplied globally via river runoff.

Simulating both surface (primary production, grazing, egestion and excretion by zooplankton) as well as deep (sinking and decay of organic matter) processes before the background of ocean circulation and seasonally varying forcing, the model thus

encompasses processes that act on a variety of time scales, from the order of hours to days (surface) to months and years.

## 2.2 Optimization

### 2.2.1 The optimization algorithm CMA-ES

The TMM as described above is fast enough to be used together with meta-heuristic methods for parameter optimization, such as Evolutionary Algorithms (EAs) or Estimation of Distribution Algorithms (EDAs). Although these methods require more

function evaluations to converge to some local optimum than gradient-based methods, they are of advantage in complicated, irregular "search landscapes" with local optima (which might be far worse than the global optimum), or discontinuities.

The common goal of such population-based meta heuristics is to strike a good balance of both search properties, exploration and exploitation. Classical evolutionary algorithms as depicted on the left of Fig. 1 mimic principles of natural evolution to pursue that goal. They use randomized procedures to select, combine, mutate and reinsert candidate solutions (individuals)

from/into a given solution set (population). In each iteration, these mechanisms (red operations in Fig. 1) indirectly imply a probability distribution on the search space with respect to which individuals are likely to appear in the next generation. The implied probability distribution changes in each generation, tending to increase the probabilities of good solutions and to decrease the probabilities of poor solutions due to the survival-of-the-fittest principle.

In opposite to classical EAs, estimation of distribution algorithms (sketched on the right of Fig. 1) use an explicit (pa-

rameterized) probability distribution from which candidate solutions are sampled, directly. In each iteration, the probability distribution is also updated directly by utilizing good solutions of the current iteration. Good solutions of preceding iterations are (optionally) considered by involving preceding probability distributions into the update process using auxiliary variables. Evolutionary frameworks use operators (EAs) and probability distributions (EDAs) that are appropriate for the searchspace





under consideration. For example, so called quantum inspired evolutionary algorithms (QiEA) have shown to be very suitable EDAs for binary problems (e.g. Kliemann et al., 2013; Patvardhan et al., 2015, 2016). QiEA versions for continuous problems have also been investigated in the literature.

We here use a state-of-the-art EDA for optimization of (firstly) six parameters. Our task can be classified as a continuous
optimization problem with bound-constraints, i.e. boundaries for the parameters. One appropriate EA/EDA tool is the Covariance Matrix Adaption Evolution Strategy (CMA-ES;  Hansen and Ostermeier, 2001; Hansen, 2006), which has shown good performance with respect to quality and efficiency (in terms of function evaluations) in similar applications (Hansen et al., 2010).) The algorithm is invariant regarding both order preserving transformations of the objective function and rotations and translations of the search space. Invariances of a strategy justify generalizations of empirical results, which encouraged us to
choose CMA-ES for our application.

We essentially follow the description of the $(\mu/\mu_w, \lambda)$-CMA-ES in Hansen (2016). We present the guiding ideas in Subsubsections 2.2.2 - 2.2.6. For the sake of completeness, the pseudo code can be found in 2.3. This basic version does not consider bound constraints. We therefor use a penalty function based boundary handling (Hansen et al., 2009) which we will briefly explain in Subsubsection 2.2.7

### 2.2.2 Normal distributions

In CMA-ES the distribution from which candidate solutions (BGC parameter vectors in our application) are sampled is a multi-variate normal-distribution. It generalizes the usual normal distribution, also known as Gaussian distribution or Gaussian bell, from $\mathbb{R}$ to the vector space $\mathbb{R}^n$ with arbitrary dimension $n$, given by the number of biogeochemical parameters to be estimated. The position and the (bell)shape of the one-dimensional normal distribution (more precisely, its density function) is
uniquely defined by its mean (the position of its top) and its variance, respectively. With respect to a given variance, the normal distribution is considered to provide the best search diversity amongst all distributions having the same variance.

An EDA that works with Gaussian distributions is supposed to carefully update both defining distribution parameters mean and variance, in order to balance its exploration and exploitation ability. This update process is illustrated in Fig. 2. The left side shows a run of the CMA-ES algorithm on a uni-variate test function (a somewhat misuse as CMA-ES is actually not suggested
to be applied with problem dimensions less than 5). The test function has many local optima in which a gradient based search might get stuck. From the Gaussian bells (the blue density functions), we draw 10 samples per iteration with function values shown as dots and update the distribution by involving the 5 better samples (blue dots). We can observe that the mean of the bell is attracted towards the good samples, then. Also, the distribution shape widens, after good samples had some distance to each other and/or some distance to the current mean. Vice versa, if all good samples are close to the mean, the shape will
narrow, again. Now, the mean of the bell is supposed to drift toward the global optimum and should then start to narrow more and more. This behavior is observed in iterations 16, 22 and 28. So, when necessary, the procedure is supposed to become less exploring but more exploiting.

Similarly to the definition of the uni-variate Gaussian distribution by mean and variance, a multi-variate normal-distribution can be uniquely identified by a mean vector $\overline{x}$ and a positive definite matrix $\mathbf{C}$ of covariances, respectively, and is denoted





by $\mathcal{N}(\overline{\boldsymbol{x}}, \mathbf{C})$. Again, the mean defines the position of the bell while the covariance matrix defines its shape. The area of one standard deviation which is an interval $[x - \sigma, x + \sigma]$ in the one-dimensional case becomes an $n$-dimensional ellipsoid, now (cf. the ellipses on the right side of Fig. 2 for $n = 2$). It can be shown that the principle axes of the ellipsoid correspond to $\mathbf{C}$'s Eigen values and Eigen vectors, respectively. More precisely, an Eigen vector defines the orientation of a principle axis and the square root of the corresponding Eigen value defines the length of that principle axis.

### 2.2.3 Sampling the distribution

Sampling a multi-variate normal distribution $\mathcal{N}(\overline{\boldsymbol{x}}, \mathbf{C})$ can be practically implemented using an Eigen decomposition $\mathbf{C} = \mathbf{B}\mathbf{D}^2\mathbf{B}^{\mathrm{T}}$, where $\mathbf{D}^2$ is a diagonal matrix of Eigen values of $\mathbf{C}$ and $\mathbf{B}$ is a matrix of corresponding orthonormal Eigen vectors of $\mathbf{C}$. One sample $\boldsymbol{x} \in \mathbb{R}^n$ of $\mathcal{N}(\overline{\boldsymbol{x}}, \mathbf{C})$ can be realized by drawing $n$ real numbers from the uni-variate standard normal distribution $\mathcal{N}(0, 1)$ to be the components of a random vector $\boldsymbol{z} \in \mathbb{R}^n$ and setting $\boldsymbol{x} = \overline{\boldsymbol{x}} + \mathbf{B}\mathbf{D}\boldsymbol{z}$.

### 2.2.4 Updating the distribution: basic principle

Empirical (re)estimates $\overline{\boldsymbol{x}}_{\mathrm{emp}}$ and $\mathbf{C}_{\mathrm{emp}}$ of the distribution parameters can be calculated from a set $S = \{\boldsymbol{x}_1, \ldots, \boldsymbol{x}_\lambda\}$ of $\lambda$ samples, such that the expectation of $\overline{\boldsymbol{x}}_{\mathrm{emp}}$ is $\overline{\boldsymbol{x}}$ and the expectation of $\mathbf{C}_{\mathrm{emp}}$ is $\mathbf{C}$. Clearly, the estimates become the more reliable the larger $\lambda$ is (like for the average score when rolling a dice many times). We may assume that the population $S$ is increasingly ordered (ranked) with respect to the considered objective function $f : \mathbb{R}^n \longrightarrow \mathbb{R}$, that is $f(\boldsymbol{x}_1) \leq f(\boldsymbol{x}_2) \cdots \leq f(\boldsymbol{x}_\lambda)$. Now, by involving only the better half of $\mu = \lfloor \frac{\lambda}{2} \rfloor$ samples, their distribution estimate $\mathcal{N}(\overline{\boldsymbol{x}}_\mu, \mathbf{C}_\mu)$ with corresponding parameters $\overline{\boldsymbol{x}}_\mu$ and $\mathbf{C}_\mu$ will be biased towards reproducing that $\mu$ samples with higher probability than the other $\lambda - \mu$ samples. CMA-ES uses values $w_1 \geq w_2 \geq \cdots \geq w_\mu$ with $\sum_{i=1}^n w_i = 1$ to give solutions a rank dependent weight in the updating process of both, $\overline{\boldsymbol{x}}_\mu$ and $\mathbf{C}_\mu$ (a more general version allows to involve all solutions, applying negative weights for the poor ranks). The new mean is, thus, calculated as $\overline{\boldsymbol{x}}_\mu = \sum_{i=1}^\mu w_i \boldsymbol{x}_i$. A subtlety is the choice of the reference mean value used for estimating $\mathbf{C}_\mu$. Instead of the new empirical mean $\overline{\boldsymbol{x}}_\mu$, the mean $\overline{\boldsymbol{x}}$ of the former distribution is chosen and yields

$$\mathbf{C}_\mu = \sum_{i=1}^\mu w_i (\boldsymbol{x}_i - \overline{\boldsymbol{x}})(\boldsymbol{x}_i - \overline{\boldsymbol{x}})^{\mathrm{T}}.$$

It has the effect that the new distribution is elongated into directions of descend (see iteration 2 in the right example of Fig. 2).

### 2.2.5 Updating the distribution: reliability with small populations

As mentioned above, reliable distribution estimates require a sufficiently large number of samples. But, for a competitive computational performance we must get along with a rather small number of samples. CMA-ES therefor involves the information of former populations by updating the covariance matrix $\mathbf{C}$ to be a (convex) combination of both the current $\mathbf{C}$ and its estimate $\mathbf{C}_\mu$, that is

$$\mathbf{C} \leftarrow (1 - c_\mu)\mathbf{C} + c_\mu \mathbf{C}_\mu. \tag{1}$$





Using this formula, it can be shown that $37\%$ of the current matrix $\mathbf{C}$'s information dates back at least $\lfloor \frac{1}{c_\mu} \rfloor$ generations, that is, the choice of the smoothing factor $c_\mu$ decides about the backward time horizon of the update procedure.

Another feature that facilitates small population sizes $\lambda$ is to calculate and update a vector $\boldsymbol{p}_c$ that represents iteration averaged changes of the distribution mean and to use $\boldsymbol{p}_c$ for a so called rank-one estimate $\mathbf{C_1} = \boldsymbol{p}_c \boldsymbol{p}_c^{\mathrm{T}}$ of the covariance matrix.

The idea behind this approach is that, using $\mathbf{C}_\mu$, distribution elongations into directions of descend do not distinguish for the sign of the directions. The use of the vector $\boldsymbol{p}_c$ (called evolution path) mitigates this effect. Consecutive changes of the distribution mean into opposite directions would cancel out each other. Similar to the smoothing with factor $c_\mu$ in the update of $\mathbf{C}$, above, the update of $\boldsymbol{p}_c$ is done with a smoothing factor $c_c$. With a further smoothing factor $c_1$ for the rank-one estimate $\mathbf{C_1}$, the combined covariance matrix update reads

$$\mathbf{C} \leftarrow (1 - c_\mu - c_1)\mathbf{C} + c_\mu \mathbf{C}_\mu + c_1 \mathbf{C_1}.$$

While $\mathbf{C}_\mu$ efficiently involves information from the current population into the update process, $\mathbf{C_1}$ exploits correlations between generations. The former is important in large populations, the latter is particularly important in small populations.

### 2.2.6    Step size control

Finally, there is an additional explicit adaption of the over all scale (the step size) of the distribution by adapting a scaling
factor $\sigma$, actually using $\mathcal{N}(\overline{\boldsymbol{x}}, \sigma^2 \mathbf{C})$ instead of $\mathcal{N}(\overline{\boldsymbol{x}}, \mathbf{C})$. Similar to the evolution path $\boldsymbol{p}_c$ for the rank-one covariance matrix estimates above, the adaption of the scale $\sigma$ involves an evolution path $\boldsymbol{p}_\sigma$ that mirrors cumulative changes of the mean. The difference between the update formulas of both evolution paths $\boldsymbol{p}_\sigma$ and $\boldsymbol{p}_c$ is that for $\boldsymbol{p}_\sigma$ all step sizes are re-scaled with respect to the isotropic normal distribution $\mathcal{N}(0, \mathbf{I})$. The expected step size between the distribution mean of consecutive iterations is therefor the expected length of a sample of $\mathcal{N}(0, \mathbf{I})$, which is

$$\chi := \mathbb{E}\big(\|\mathcal{N}(0, \mathbf{I})\|\big) \approx \sqrt{n}(1 - \frac{1}{4n} + \frac{1}{21n^2}).$$

Now, a rather small length $\|\boldsymbol{p}_\sigma\|$ compared to $\chi$ indicates that consecutive normalized moves of the mean canceled each other out, meaning that the overall scale of the distribution should be reduced with $\sigma$. Vice versa, an evolution path $\boldsymbol{p}_\sigma$ longer than $\chi$ indicates consecutive distribution drifts into correlated directions which justifies a larger overall scale of the distribution.

### 2.2.7    Boundary handling

In order to consider boundary constraints we use the procedure proposed in Hansen et al. (2009, Section IV B) for CMA-ES. It applies if the distribution mean runs out of bounds. In this case, the objective function value of an infeasible sample $\boldsymbol{x}$ becomes the sum of the fitness of its closest feasible point $\boldsymbol{x}_{\text{feas}}$ and a weighted quadratic penalty function of its distance $\|\boldsymbol{x} - \boldsymbol{x}_{\text{feas}}\|$ to the feasible box (to $\boldsymbol{x}_{\text{feas}}$). Thus, feasible samples are never penalized and the minimum of the penalized fitness function lies within the feasible box. The quadratic penalty function has coordinate-wise weights $\frac{\gamma_i}{\xi_i}$, where $\xi_i$ scales the out of bounds
distance in the $i$-th coordinate with regard to the shape of the current distribution. The $\gamma_i$ are suitably initialized with the range





of former (unpenalized) objective function values and is multiplied with a constant $> 1$ in every iteration in which $\overline{x}_i$ is more than 3 standard deviations off its bounds.

In our implementation of CMA-ES, the feasible box we operate on is the unit cube $[0,1]^n \subseteq \mathbb{R}^n$. The samples are then linearly transformed (encoded) with respect to the actual bound constraints before evaluating the objective (misfit) function.

## 2.3 Implementation of the optimization algorithm

### 2.3.1 Algorithm outline

The CMA-ES approach described in Subsection 2.2.1 allows for reliable covariance matrix estimates with a relatively small population size. The default population size of $\lambda = 4 + 3\log(n)$ individuals and all further operational constants are successively derived from the problem dimension $n$ as outlined in Table 1.

Here, $\mu$ counts the good portion of individuals that are selected from the $\lambda$ samples in each iteration and used to update the probability distribution. As mentioned in Subsubsection 2.2.4, sampled individuals are always sorted with respect to their function values $(f(\boldsymbol{x_1}) \leq \cdots \leq f(\boldsymbol{x_\lambda}))$. The $\mu$ recombination weights $w_i$ sum up to 1 and are monotonically decreasing in order to give better selected samples a higher weight in the updating formulas. Our present setting of the weights corresponds with the MATLAB example code in Hansen (2016) but differs from the improved setting that has been newly introduced in that work. The value $\mu_{\text{eff}}$ depends on the choice of the weights and lies between 1 and $\mu$ if the weights sum up to 1. Together with the problem dimension $n$, it appears in the calculation of the four smoothing constants $c_\sigma, c_c, c_\mu, c_1$ used in the update formulas of both the evolution paths and the covariance matrix. Their dependence on $n$ and $\mu_{\text{eff}}$ have been derived empirically. The formula for the damping parameter $d_\sigma$ differs from the original one but yields the same value for the weights we choose. The constant $\chi$ (cf. Subsubsection 2.2.6) is approximately the expected norm of the $n$-dimensional standard normal distribution $\mathcal{N}(0, \mathbf{I})$.

The algorithm details are summarized in Algorithm 1. It starts with the identity matrix $\mathbf{I}$ for the covariances, that is, with an isotropic distribution. Assuming the optimum solution to reside within the unit cube $[0,1]^n \subseteq \mathbb{R}^n$, the mean $\overline{x}$ and the over all scale $\sigma$ are initialized according to Hansen (2016). Actually, having bound constraints (cf. Subsubsection 2.2.7) we operate on the unit cube and shift and scale obtained samples into their real bounds before calculating their objective function values. New samples are drawn as described in Subsubsection 2.2.3. The $\boldsymbol{y}_k$ correspond to the $\boldsymbol{x}_k - \overline{\boldsymbol{x}}$ considered there, divided by the step size $\sigma$. The new $\overline{\boldsymbol{x}}$ is calculated according to $\overline{\boldsymbol{x}}_\mu$ in Subsubsection 2.2.4. Note that $\overline{\boldsymbol{y}}$ is the $\sigma$-adjusted move of the mean while $\overline{\boldsymbol{y}}^*$ adjusts the move of the mean with respect to the (isotropic) standard normal distribution. The evolution paths which cumulate the drifts of the distribution mean (adjusted with regard to the overall scale and with regard to isotropy, respectively) are updated using the corresponding smoothing factors. Here, the factors before $\overline{\boldsymbol{y}}$ and $\overline{\boldsymbol{y}}^*$ act as normalization constants (Hansen, 2016). Finally, the overall step size and the covariances are updated as described in Subsubsections 2.2.6 and 2.2.5, respectively. We stop either after the predefined number of iterations or if the current population shows a flat misfit distribution, i.e., if the fitness of the better 70% of the individuals deviate less than $\epsilon = 10^{-5}$ from the very best one.





---

**Algorithm 1** The $(\mu/\mu_w, \lambda)$-CMA-ES

---

**Initialization:**

Set $\lambda, \mu, w, \mu_{\text{eff}}, \chi, c_\sigma, d_\sigma, c_c, c_\mu, c_1$ according to Table 1

Set $\overline{\boldsymbol{x}} = (\frac{1}{2}, \dots, \frac{1}{2})^{\mathrm{T}}$

Set $\boldsymbol{p}_\sigma = \boldsymbol{p}_c = 0, \ \mathbf{C} = \mathbf{B} = \mathbf{D} = \mathbf{I}$ and $\sigma = 0.5$

**while** stopping criterion is not met **do**

   **Sample probability distribution:**

      **for** $k = 1, \dots, \lambda$ **do**

         Sample $\boldsymbol{z}_k \in \mathbb{R}^n$ from $\mathcal{N}(0, \mathbf{I})$ by sampling its entries from $\mathcal{N}(0, 1)$

         Set $\boldsymbol{y}_k = \mathbf{B}\mathbf{D}\boldsymbol{z}_k$ and $\boldsymbol{x}_k = \overline{\boldsymbol{x}} + \sigma \boldsymbol{y}_k$

      **end for**

   **Update probability distribution:**

      Update mean:

         $\overline{\boldsymbol{x}} \leftarrow \sum_{k=1}^{\mu} w_k \boldsymbol{x}_k$

         Set $\overline{\boldsymbol{y}} = \sum_{k=1}^{\mu} w_k \boldsymbol{y}_k$ and $\overline{\boldsymbol{y}}^* = \mathbf{B}\mathbf{D}^{-1}\mathbf{B}^{\mathrm{T}}\overline{\boldsymbol{y}}$

      Update evolution paths:

         $\boldsymbol{p}_\sigma \leftarrow (1 - c_\sigma)\boldsymbol{p}_\sigma + \sqrt{c_\sigma(2 - c_\sigma)\mu_{\text{eff}}}\, \overline{\boldsymbol{y}}^*$

         $\boldsymbol{p}_c \leftarrow (1 - c_c)\boldsymbol{p}_c + \sqrt{c_c(2 - c_c)\mu_{\text{eff}}}\, \overline{\boldsymbol{y}}$

      Update covariances and scaling:

         $\sigma \leftarrow \sigma \cdot \exp\left(\frac{c_\sigma}{d_\sigma}\left(\frac{\|\boldsymbol{p}_\sigma\|}{\chi} - 1\right)\right)$

         Set $\mathbf{C}_\mu = \sum_{k=1}^{\mu} w_k \boldsymbol{y}_k \boldsymbol{y}_k^{\mathrm{T}}$ and $\mathbf{C_1} = \boldsymbol{p}_c \boldsymbol{p}_c^{\mathrm{T}}$

         $\mathbf{C} \leftarrow (1 - c_\mu - c_1)\mathbf{C} + c_1 \mathbf{C_1} + c_\mu \mathbf{C}_\mu$

         Determine $\mathbf{B}$ and $\mathbf{D}$ from Eigen decomposition $\mathbf{C} = \mathbf{B}\mathbf{D}^2\mathbf{B}^{\mathrm{T}}$

**end while**

---

### 2.3.2 Algorithm parallelization

Our current technical implementation of the parallel framework can be easily transferred to other EAs/EDAs. The iterative optimization process is carried out via a series of chain jobs, where short serial jobs (the actual optimizer) that update the population of model evaluations ("individuals"; i.e. parameter sets for biogeochemistry) alternate with parallel jobs of function

5   evaluations ("generations"), i.e. forward integrations of the coupled ocean model with different parameter sets. Parameters of the optimizer are population size $\lambda$ and the termination criterion for convergence, additionally a maximum number of iterations.

As noted above, the framework presented here is set up such that a serial script `serial.job` calls the optimization routine (in our case CMA-ES), which computes a population of size = $\lambda$ of parameter vectors, stored in ASCII files. The same script then calls a parallel script `parallel.job`, which starts $\lambda$ model simulations. During these simulations, the parameter files are

10   read, and a spinup is carried out for each individual setup. The individual model runs then output the misfit function to specified files. When all jobs are finished, script `parallel.job` invokes script `serial.job` again, etc.. Thus, communication





between both alternating steps (creation of parameter vectors and computation of resulting misfit function) is carried out by these parameter and misfit files. In addition, file `nIter.txt` keeps track of the progress of optimization, and provides the information which generation is to be computed; it also contains the runtime parameters for the optimizer, CMA-ES. See information in supplement for more details on how this setup works, and how to specify biogeochemical and optimizer

parameters used e.g., in the work presented here.

## 2.4    Misfit function

As a first approach to optimization, we have calculated the root-mean-square error RMSE between simulated and observed (or twin) annual mean phosphate, nitrate, and oxygen concentrations on a global scale, weighted by the volume $V_\mathrm{i}$ of each individual grid box, expressed as fraction of total ocean volume, $V_\mathrm{T}$. To sum the three different components of the misfit

function we have to divide them by some typical value. Here we use the global mean concentration of observed tracers. The resulting misfit function $J$ thus reads:

$$J = \sum_{j=1}^{3} \frac{1}{\overline{o_j}} \sqrt{\sum_{i=1}^{N} (m_{i,j} - o_{i,j})^2 \frac{V_i}{V_\mathrm{T}}} \tag{2}$$

for the annual mean concentrations of three tracers phosphate ($j = 1$), nitrate ($j = 2$) and oxygen ($j = 3$), at $N = 52749$ locations (model grid boxes) of the model domain. $\overline{o_j}$ is the global average observed (or twin) concentration of the respective

tracer. $m_{i,j}$ and $o_{i,j}$ are model and observations (or twin results), respectively. By weighting the model mismatch with volume, we put some emphasis on the deep ocean, down-weighting deviations in surface grid boxes relative to those of deep boxes. Thus, our misfit function serves more as a long time-scale geochemical estimator, in contrast to a function that focuses on (rather fast) turnover in the surface layer.

## 2.5    Parameters to be estimated

Although the model contains more than 20 parameters (even more, if we consider the empirically derived parameters for benthic burial, nitrogen fixation, denitrification and air-sea gas exchange; see Kriest and Oschlies, 2013, 2015), for this first approach we only consider six parameters for optimization. Four parameters are more relevant for biological interactions at the sea surface. Phytoplankton growth is controlled by the half-saturation for light ($I_\mathrm{c}$, in W m$^{-2}$) and phosphate ($K_\mathrm{PHY}$, in mmol P m$^{-3}$). For optimization of zooplankton parameters we chose its maximum grazing rate ($\mu_\mathrm{ZOO}$, in d$^{-1}$) and quadratic

mortality rate ($\kappa_\mathrm{ZOO}$, in (mmol P m$^{-3}$)$^{-1}$ d$^{-1}$). Two parameters are of importance for the transport and decay of particulate organic matter to/in the deep ocean, namely the ratio of oxygen consumption to phosphate release during aerobic remineralization ($R_{-\mathrm{O2:P}}$, mmol O$_2$:mmol P), and the parameter for vertical increase of sinking speed of organic matter, $a$ (d$^{-1}$). Note that as stated above, in the following, and during optimization, we express this last parameter through $b = r/a$, with $r$ held constant at $r = 0.05$ d$^{-1}$.



## 2.6 Setup and performance of optimization

Using the combined framework described above, i.e. TMM+MOPS+CMAES, we carried out five different, full optimizations: one against annual average phosphate, nitrate and oxygen of year 3000, simulated by an experiment that applies the same biogeochemical parameters as MOPS-RemHigh of Kriest and Oschlies (2015), setup "base" (i.e., with a particle flux described by $b = 0.858$, or $a = 0.058275$). We refer to this experiment as "TWIN". Four further optimizations were carried out against observations of annual mean phosphate, nitrate, and oxygen (Garcia et al., 2006a, b), gridded onto the model geometry. These are referred to as OBS-WIDE, OBS-WIDE-20, OBS-NARR and OBS-NARR-R.

To fully explore the capabilities of the CMAES, for experiment TWIN, OBS-WIDE and OBS-WIDE-20 we first set rather wide boundary constraints (parameter boundaries; see table 2). The second set of optimizations against observations was carried out with a narrower range of zooplankton parameters (OBS-NARR). In this latter experiment, the boundaries for zooplankton parameters are restricted to $\pm 50\%$ of the value of the reference run of MOPS. We finally evaluate the robustness of optimization OBS-NARR by repeating this optimization with a different random selection of the parameters from the distribution calculated by CMAES (experiment OBS-NARR-R).

Four of the five optimizations were carried out using a population size $\lambda$ of 10, which was deemed sufficient for six parameters, given the default configuration of the CMAES (see above). To investigate more closely a potential local minimum that occurred in OBS-WIDE, in experiment OBS-WIDE-20 we increased the population size to $\lambda = 20$.

The internal termination criterion of CMAES was reached after 95, 173, 182 and 140 generations for OBS-WIDE, OBS-WIDE-20, OBS-NARR and OBS-NARR-R, respectively. For the twin experiment, we restricted the maximum number of generations to 200, at which TWIN had approached the target parameters, the misfit declined to $< 0.0004$ (i.e., on average less that $0.2‰$ of global mean tracer concentrations; see Eqn. 2) and fitness variance declined to $< 10^{-9}$. As presented above, in each "generation" we computed 10 (20) different "individuals" (model simulations over 3000 years) in parallel. One simulation of each generation on average took $\approx 1.25$ hours, on 40 (80) nodes of Intel Xeon IvyBridge or Intel Xeon Haswell at the North-German Supercomputing Alliance (HLRN). We note that tests on either hardware (two iterations of the coupled code, started from generation 80 and 160 of experiment TWIN) did not reveal any differences in the estimated fitness. The CMAES - which, due to its very short runtime, is not parallelized - was always computed on one core of Intel Xeon IvyBridge.

# 3 Results

## 3.1 Twin experiment (TWIN)

The optimization starts with a wide range of potential parameters (see Fig. 3), with individual parameters sometimes even exceeding the prescribed boundaries. This results in high maximum and minimum misfit (Fig. 4), and this high variability is maintained over about 10-20 generations. The trajectory of transient average parameter values and their variance depend strongly on the parameter itself: while the two parameters associated with rather long time scales and large ocean volumes, namely the stoichiometric ratio $R_{-O2:P}$ and exponent $b$ describing particle sinking, approach their target values quite early





(about generation 20-40), parameters associated with surface biogeochemistry stay far away from their target value for $\approx 80$ generations ($I_c$, $K_{PHY}$, $\kappa_{ZOO}$), or oscillate around it ($\mu_{ZOO}$). After $\approx 160$ generations, most of the parameters reached their target value, the exception being the half-saturation constant of phytoplankton for phosphate uptake, $K_{PHY}$ (Table 3). This parameter still shows considerable variability at the end of the optimization (generation 200), although by that time is it quite

close to the - rather low - target value.

The misfit function, its variance and the parameter variance do not decrease monotonously throughout the optimization trajectory. In particular, after an initial decline over ca. 60 generations, parameter and misfit variance increase again. Further increases in variance can be seen around generation 100, and at the end, when the algorithm widens its search area again, probably in search for an optimal $K_{PHY}$. It seems encouraging that the algorithm obviously does not get stuck in a local

minimum, but, at the expense of deterioration of the misfit, continues to search for an even better parameter set.

The largest fraction of the misfit function is related to oxygen, followed by the misfit to nitrate, and then phosphate. The dominance of oxygen and nitrate is not surprising, as these tracers are not conservative; i.e., their global inventory might change due to air-sea gas exchange, denitrification and nitrogen fixation (see also Kriest and Oschlies, 2015), so that the model may not only err with respect to the spatial distribution of these tracers, but also with respect to their global mean concentration.

In Fig. 5 we finally exploit the shape of the misfit function, shown on a color scale for each two pairs of parameters. As can be seen from misfit plotted against $R_{-O2:P}$ and $b$ (upper right corner), these two parameters are quite well constrained, with a very well defined minimum around the target value. All other parameters show more or less elongated search "canyons". Much of the algorithm search starts away from the target value; however, the algorithm finally manages to approach the target value even when the search path is not straight, but curved in the two-dimensional projections of the parameter space. Further, even

when the algorithm exceeds the target value (e.g., for the maximum growth rate of zooplankton, $\mu_{ZOO}$; lower right corner), despite of the already low misfit function the algorithm finally returns to the somewhat lower value (compare also to Fig. 3, lower left panel).

Summarizing, CMAES seems capable to deal even with our irregular search landscape, when iterated for a long enough time and with a sufficiently large population size. Some problem remains with regards to the half-saturation constant of phytoplank-

ton for phosphate uptake: zooming into the scatter plot presented in Fig. 5 reveals that for this parameter the search landscape becomes quite uninformative (Fig. 6), with similar misfits close to the optimum. One reason for this low sensitivity of the misfit function may be found in the fact, that in the twin, against which the model is optimized, only very few (1%) phosphate values are at or below the target value of $K_{PHY} = 0.03125$ mmol P m$^{-3}$. Therefore, besides the dominance of oxygen in the misfit function (Fig. 4) the misfit function is further dominated by phosphate concentrations outside the oligotrophic surface regions,

rendering it quite insensitive to changes in the half-saturation constant at low values.

### 3.2 Optimization against observed nutrients and oxygen distributions

#### 3.2.1 Wide boundary constraints for zooplankton (OBS-WIDE, OBS-WIDE-20)

When optimizing the model against observed concentrations with exactly the same setup as for experiment TWIN, optimization OBS-WIDE reaches the internal termination criterion of the CMAES at generation 95. Instead of declining exponentially

towards zero, the misfit only declines from an average initial value of $\approx 0.8$ to 0.477 (Fig. 7, Table 3), i.e. only slightly less than the misfit of the reference run (0.529). Also, the variance of misfit, as well as that of the parameters show a more or less gradual decline, without any intermittent increase (see supplement). Another notable difference to TWIN is the higher contribution of phosphate to the misfit function (Fig. 7).

Some parameters diverge strongly from those of there reference run. In particular, the phytoplankton's half-saturation con-

stant for light, $I_c$, increases strongly up to its upper boundary (Fig. 8; Table 3; see also supplement for a plot of topography of the misfit function). However, the stronger light-limitation of phytoplankton growth is counteracted by a strong decrease in zooplankton growth rate, $\mu_{ZOO}$, and a strong increase in its quadratic mortality rate, $\kappa_{ZOO}$. As a consequence, average and maximum zooplankton concentrations are $< 25\%$ and $< 50\%$ of that of the reference run in the surface layer (Fig. 9), while phytoplankton is strongly increased, when compared to the reference run. Most likely because the zooplankton-detritus

pathway is nearly shut off, DOM concentrations are strongly increased. The reorganization of the pelagic food web in this optimized model scenario is reflected in the global annual biogeochemical fluxes: primary production is enhanced by almost 14%, but loss through grazing is reduced to about 1/3 of that of the reference run (Table 4). As a consequence, the largest fraction of recycling is through remineralization of detritus and DOM ($> 95\%$ of annual production), and only 4% through zooplankton excretion, while in the reference run zooplankton recycles almost 15% of annual production. Due to the reduced

particle sinking speed shallow (130 m) and deep (2030 m) particle flux are reduced, as is benthic burial. While some of the simulated fluxes are within the observed estimates, too low zooplankton concentration, as well as resulting low zooplankton grazing are far outside observed estimates (see Table 4).

Therefore, although optimization OBS-WIDE against observations has decreased the misfit to observations to $\approx 90\%$ of that of the (subjectively tuned) reference run, the outcome is not overly satisfying with respect to the optimized parameters and

the resulting dynamical behavior of the model. Obviously, the very wide boundary constraints we chose for the zooplankton parameters led to a solution where zooplankton is almost dead - a phenomenon that does not occur in the real ocean.

To examine if this optimization became trapped in a local minimum, in experiment OBS-WIDE-20 we increased the population size of CMAES from $\lambda = 10$ to $\lambda = 20$. Due to a larger population, in this optimization the variability of fitness (Fig. 10) and parameter values (Fig. 11) is maintained over a longer period, again, as for optimization TWIN, with intermittent increases

of variance during the course of the optimization. Most importantly, using the setup of OBS-WIDE-20 the optimization finds very different parameters for many of the biogeochemical components:

$R_{-O2:P}$ is now closer to the a priori value of 170, while optimal $b$ has increased considerably to $b = 1.34$ (Table 3). The largest difference to both the reference run as well as optimization OBS-WIDE occurs for the four biogeochemical parameters that are more closely tied to surface processes: $I_c$ decreases to less than 50% of its a priori value, while $K_{PHY}$ is at its upper



boundary of $0.5\,\mathrm{mmol\,P\,m^{-3}}$. Encouragingly, zooplankton parameters are now such that zooplankton is viable (Fig 9). Its maximum growth rate is very close to the a priori value of $2\,\mathrm{d^{-1}}$. Its mortality rate is still quite high; however, because of its high growth rate zooplankton plays a considerable role in the pelagic nitrogen budget , with global fluxes much closer to the observed ones than for optimization OBS-WIDE (Table 4).

Summarizing, using a larger population size and thus a denser scan of the parameter space (see Fig. 12), CMAES has found a better solution, with respect to the misfit function (see Table 3) as well as a closer fit to biogeochemical fluxes and more plausible biological patterns.

### 3.2.2    Narrow boundary constraints for zooplankton (OBS-NARR and OBS-NARR-R)

Optimizations with a population size of $\lambda = 20$, as for OBS-WIDE-20, are computationally quite expensive, especially when
iterated over a large number of generations (Table 3). Via the quite wide boundary constraints for zooplankton parameters, we have assumed to have almost no knowledge about zooplankton. In the following two sensitivity experiments we examine the impact of this assumption on optimization performance, by restricting zooplankton parameters to a narrower range. These experiments are again carried out with a population size of $\lambda = 10$.

To enforce live zooplankton, we restricted the range of zooplankton parameters to $\pm 50\%$ of their reference value. This results
indeed in a solution with organic tracer concentrations close to that of the reference run or OBS-WIDE-20 (Fig. 9). After 182 generations, the algorithm terminates with a misfit of 0.45 (Fig. 13), i.e. better than experiment OBS-WIDE, but the same as for optimization OBS-WIDE-20 (Table 3). As in TWIN and OBS-WIDE-20, misfit variance shows intermittent increases, and the contribution of nitrate to the misfit function dominates over that of phosphate. Likewise, resulting optimal parameter values are quite close to those of OBS-WIDE-20 (Table 3). Thus, OBS-NARROW is more similar to OBS-WIDE-20 than to
OBS-WIDE, demonstrating the importance of good a priori knowledge about parameter values.

As for OBS-WIDE-20, the quadratic mortality of zooplankton, $\kappa_{\mathrm{ZOO}}$ and the half-saturation constant of phosphate uptake for phytoplankton, $K_{\mathrm{PHY}}$ show a strong increase; the latter up to its upper prescribed boundary, which may be interpreted as an attempt of the algorithm to force the model towards higher surface nutrient concentrations in the subtropical gyres. A reduced half-saturation constant for light, on the other hand, counteracts the grazing pressure exerted by zooplankton, particularly in
the high latitudes. Most likely because of increased detritus production by zooplankton - and thus increased export from the surface layer (Table 4) - particle flux to the deep ocean is reduced by an increase in $b$, i.e. relatively slow particle sinking speed.

A closer look at the topography of the misfit function shows that for some parameters it is quite insensitive to changes (Fig. 15; see supplement for a detailed plot of misfit topography around $\pm 2\%$ of the optimal parameters). While again the parameters $R_{-\mathrm{O2:P}}$ and $b$, that tend to exert an influence on large temporal and spatial scales, are quite well constrained, many
of the surface-related parameters, that act on smaller time scales, such as $K_{\mathrm{PHY}}$, show a wide scatter across the parameter space, with very little differences in the misfit function.

However, variations in parameters after $\approx 40$ generations do not strongly improve the model fit to observations (Figures 13 and 14). The rather constant misfit after generation 40 is quite surprising, given that some parameters still show some significant excursions after that time, indicating that - as already shown in Fig.15 - the misfit function is quite uninformative about these





parameters. This insensitivity of abiotic tracers is also illustrated in Fig. 16, which shows the deviation of vertically integrated tracers from observations, plotted for individuals of three different generations of OBS-NARR (see also blue vertical lines in Fig. 14) The parameters of these individuals differ mainly with respect to their combination of $K_{\mathrm{PHY}}$ and $\kappa_{\mathrm{ZOO}}$. While the reference run applies very low $K_{\mathrm{PHY}} = 0.03125$ mmol P m$^{-3}$ and moderate $\kappa_{\mathrm{ZOO}} = 3.2$ (mmol P m$^{-3}$)$^{-1}$ d$^{-1}$, individuals

of the optimization are characterized by medium (generation 61) to high (generation 110 and 182) $K_{\mathrm{PHY}}$, and moderate (generation 110), slightly increased (generation 61) and high (generation 182) $\kappa_{\mathrm{ZOO}}$ (see also blue vertical lines in Fig. 14). All individuals differ from the reference run; yet the difference among them is almost not visible in the simulated tracer distributions. Thus, annual mean tracer concentrations on a global scale do not seem to suffice in constraining some of the parameters related to the very dynamic biological turnover at the sea surface.

Except for deep particle fluxes, all biogeochemical fluxes are increased compared to the reference run or experiment OBS-WIDE, but similar to that of OBS-WIDE-20 (Table 4). Therefore, although the misfit function so far only optimized towards inorganic constituents, the optimized model with narrow zooplankton parameter boundaries shows a much better fit to observed global fluxes to primary production, zooplankton grazing, shallow and deep particle flux, and benthic burial. The seemingly better dynamical biogeochemical behavior of this model setup gives some confidence that the model's fit to inorganic tracers

is not improved on cost of any other tracer.

Repeating optimization OBS-NARR with a different random selection of parameters from the parameter distribution in each generation (OBS-NARR-R) yields the same, or very similar, best values for most of the parameters (see Table 2), the exception being the two zooplankton parameters, $\mu_{\mathrm{ZOO}}$ and $\kappa_{\mathrm{ZOO}}$. These two parameters of OBS-NARR-R are 7% ($\mu_{\mathrm{ZOO}}$) and 16% ($\kappa_{\mathrm{ZOO}}$) lower than in OBS-NARR; however, the misfit of both optimizations is the same (0.45). The low sensitivity of the

misfit function to zooplankton parameters is mirrored in similar nutrient and oxygen distributions (see supplement) and almost identical biogeochemical fluxes (see Table 4).

## 4 Discussion

### 4.1 Computational performance

Our results suggest that the CMAES optimization algorithm performs well, particularly for the twin experiment, even though

the parameters to be estimated involve diverse temporal and spatial scales. CMAES manages to set up curved search paths in parameter space, and therefore is capable to approach an optimum within a rather complex topography of the misfit function. Its sometimes elongated and/or curved shape resembles many of those resulting from earlier 1D (Athias et al., 2000; Schartau et al., 2001; Schartau and Oschlies, 2003a; Ward, 2009) or 3D (Kwon and Primeau, 2006, 2008) optimizations of marine biogeochemical models. However, when imposing wide boundary constraints for zooplankton parameters, OBS-WIDE becomes

trapped in a local minimum; only with a larger population size or narrower parameter boundaries we find a solution that results in realistic concentrations and fluxes of all components. Clearly, the number of experiments conducted here is too small to make statistically significant statements about the optimizers' exploration capability with respect to the population size. But similar to other population based heuristics, examinations with multimodal test functions have given evidence that larger pop-





ulations increase CMAES' chance to find good local optima (or even a global optimum; Hansen and Kern, 2004). It remains to be investigated, whether different configurations of the CMAES, or a different optimization algorithm, e.g., gradient-based methods or evolutionary algorithms, perform better or worse with respect to the number of model evaluations required, or their ability to avoid local minima (see also Athias et al., 2000). However, there is some indication that genetic algorithms perform

better with respect to a rough topography of the misfit function, when compared to a variational adjoint method, with otherwise equally good fit to marine biogeochemical observations (Ward et al., 2010).

As the computational effort remains a challenge in parameter optimization of global ocean BGC models, further possibilities to accelerate model evaluations within the optimization process are desirable. Surrogate-assisted approaches use meta-models to approximate model evaluations within optimization (Priess et al., 2013). They are becoming practice within evolutionary

frameworks coping with computational expensive model functions (Jin, 2011). It should be worth considering surrogate approaches with CMAES as investigated in Kern et al. (2006), Auger et al. (2013) and Loshchilov et al. (2012). A general approach with EA and EDA frameworks is to prematurely abort the fitness calculation after detecting that the corresponding individual will not be better than the worst member of the current population. We can benefit from such short-cut fitness computation if the optimizers' implementation supports asynchronous communication. An example for this approach is dealt with

in Kliemann et al. (2013). There, aborting fitness calculations reduces the computational effort by orders of magnitude, since the considered combinatorial problem is of minimax-type. However, short-cut fitness computation concerning ocean models requires a more elaborated method and is not expected to reach similar savings.

## 4.2  Misfit function and parameter identifiability

In our study we chose annual means of dissolved nutrients and oxygen on a rather coarse spatial grid as a measure for model

skill. By doing so, we avoid problems associated with time lags (e.g., in phytoplankton blooms, which would result in time lags of nutrient depletion) or meso- and submesoscale spatial structures (see, e.g., Wallhead et al., 2006), obviously on the cost of precisely resolving parameters related to the biological system in surface layers. Possibly as a consequence of this particular misfit function, the parameters that could be fitted best are parameters that are mostly influential in determining the nutrient or oxygen distribution on large spatial and temporal scales, such as the stoichiometric ratio between oxygen and phosphorus,

$R_{-\mathrm{O2:P}}$, or the parameter that determines particle sinking speed, $b$ (see also Kriest et al., 2012). Our model optimizations against observations so far confirm a stoichiometry of $R_{-\mathrm{O2:P}} \approx 170$ mmol $\mathrm{O_2}$:mmol P, in agreement with observational estimates (Takahashi et al., 1985; Anderson and Sarmiento, 1994), but suggest an increase of $b$ towards $\approx 1.3$. The latter is to some extent in agreement with results obtained by Kwon and Primeau (2006, 2008), who found an optimal $b$ of 1, when fitting a simple global model against observed inorganic tracers. It should be kept in mind, however, that the $b$ obtained in our study

resembles not only particle sinking speed, but also accounts for the effect of numerical diffusion in our rather coarse vertical grid (Kriest and Oschlies, 2011). Accordingly, the "true" $b$ can be regarded as being about 10-20% smaller than obtained by our study (manuscript in progress). Also, as has been shown earlier (Kriest and Oschlies, 2013), the lower boundary condition simulated by benthic exchange can be very important for the ability of phosphate and oxygen to constrain particle sinking; therefore, the results obtained in our study should be regarded as specific to this particular biogeochemical model.





Our optimizations against observations with wide and narrow boundaries for zooplankton parameters produced two solutions with quite similar misfit, but with very different biological parameters, and consequently different fluxes and concentrations of organic components in the surface layers. Using wide boundary constraints for zooplankton parameters resulted in a solution where zooplankton is almost extinct, while phytoplankton and DOM concentration are far too high. Solutions of optimizations

with unrealistic parameter values or concentrations for zooplankton have been observed earlier (Schartau et al., 2001; Ward et al., 2010), and point towards a necessity to better constrain this compartment. Increasing the population size $\lambda$ of CMAES in optimization OBS-WIDE-20 could cure this problem, but on the cost of a high computational demand. Restricting the range of zooplankton parameters resulted in a better fit to nutrient and oxygen; more importantly, concentrations and fluxes in the latter solution are much more realistic, confirming in the latter parameter set. This illustrates the potential benefit of a sound a

priori knowledge of parameter ranges, both in terms of biogeochemical and computational performance.

Another possibility to avoid undesired effects like nearly extinct zooplankton is to bring in further objectives which consider that issues. A technically easy approach would be to add further objective terms to the cost function. But facing complex model interactions, it can become difficult to find suitable weights for the different terms in order to force solutions to become a desired compromise of objectives. An alternative is to deal with more than one objective function, say $f_1, f_2, \ldots, f_k$. For

example, we can define the deviation of zooplankton mass from observed values as a second objective. Now, two solutions $x \neq y$ are said to be incomparable if $f_i(x) > f_i(y)$ but $f_j(x) < f_j(y)$ for some $i \neq j$. Multi-objective optimization algorithms aim to find (a limited number of) good incomparable solutions, from which the user can make a final choice that is a good compromise in his/her opinion. The topic of multi-objective optimization is intensively regarded with EAs (Deb, 2009) and EDAs (Hauschild and Pelikan, 2011), including CMAES (Igel et al., 2007).

Nevertheless, even for the more realistic optimizations OBS-WIDE-20, OBS-NARR and OBS-NARR-R we find similar misfits for a rather wide range of some phyto- and zooplankton parameters, pointing towards an indeterminacy of these parameters when using the current misfit function. These problems were also encountered by Kwon and Primeau (2006), when optimizing $b$, DOP production and its decay rate against phosphate on a global scale. They found that phosphate data alone were not sufficient to resolve parameters associated with DOP, but several equally good fits could be obtained with different

sets of parameters. It remains to be investigated, whether this is related to the lack of temporal solution, or to phosphate not being too tightly related to dissolved or particular organic matter. Subsequent studies with different misfit functions, that for example resolve monthly changes, target at the representation of surface nutrients (e.g., by using a weighted, relative misfit; Kriest et al., 2010) or add additional tracers to the misfit function (e.g., combining chlorophyll derived from remote sensing with nitrate observations; see also Tjiputra et al., 2007) will reveal the effect of the assumptions made for the misfit function

with respect to constraining these parameters.

### 4.3  Future directions

Even the use of observations more closely related to surface biology may not resolve the problem of indeterminacy, as shown by Ward et al. (2010) in optimizations of two different, 0D-biogeochemical models. As in earlier, 0D and 3D studies (e.g., Friedrichs, 2001; Schartau et al., 2001; Kwon and Primeau, 2006, 2008), they found almost identical misfits for a wide range



of parameters, an indication that these models are underdetermined, particularly when attempting to estimate more than about 10 parameters. In our study we have chosen to tune a rather moderate number of six parameters, but already noted some difficulty in constraining two of these. A potential solution could be to fix certain parameters to prior values, and thereby decrease the dimension of the parameter space to be estimated. However, as pointed out by Ward et al. (2010), this may lead

to an underestimate of model uncertainty, and therefore not be the ultimate cure for this problem. Future studies will address these problems by testing different combinations of parameters, in conjunction with different misfit functions.

The above mentioned problems may even increase if we move towards more sparsely sampled, biased, or noisy data. So far, for the twin experiment as well as for the optimization against observations we assume perfect data coverage. However, sparse data sets (as usually available from cruises or time series stations) as well as the influence of noise have been shown to be very

influential for the ability of an optimization to recover results from 0D (Friedrichs, 2001; Schartau et al., 2001; Löptien and Dietze, 2015) and 3D (Tjiputra et al., 2007) twin experiments. Future studies will have to address to what extent noise will affect the 3D optimizations presented here.

While we found a decrease of the twin experiment's misfit to almost zero, the misfit of the optimization against observations remained relatively high (on average, about 15% of global mean tracer concentrations). Potential reasons for this are an

inappropriate biogeochemical model structure, wrong choice of parameters to be optimized, or flaws in the physical model. For example, it is well known that coarse resolution models do not resolve physical processes of the Equatorial Pacific current system (Dietze and Loeptien, 2013), which may result in an attempt of the optimization to "cure" deficient physics by changing biogeochemical parameters. This feature might also explain some of the sensitivities - or lack of - found by Kwon and Primeau (2006). Solutions to this potential flaw could be to exclude regions from the misfit, that are known to be not well represented

by the physical model, or to weigh biogeochemical misfits by the model's fit to observations of physical data.

To summarize, any global model study that aims to inversely determine parameters of a global biogeochemical ocean model in an attempt to find the model setup "best" suited for a particular application (and circulation), has to consider five tasks: (1) investigate model solutions on the appropriate (depending on tunable parameters) time scales, possibly including long, millennial simulations; (2) address the potential of local minima (depending on the topography of the misfit function); (3) investigate

different parameter combinations and boundaries, including the misfit function's sensitivity to them; (4) disentangle the effects of physical and biogeochemical model on model-data misfit; and (5) investigate the effect of misfit function, including data distribution and availability on model assessment. This last point also includes decisions about weights applied to different data sets, or for a particular form of misfit function, which may be very influential for the optimal parameter choice (Evans, 2003). It also depends on the desired application of the model, and the scientific question it is supposed to address.

**5    Conclusions**

We have presented a framework for the optimization of global biogeochemical ocean models, that combines an offline approach for transport of biogeochemical tracers with an Estimation of Distribution Algorithm (Covariance Matrix Adaption Evolution Strategy, CMAES). A twin experiment revealed a good performance of this algorithm with respect to recovering six parameters,



that are associated with various time and space scales. Further tests with different setups of the optimization algorithm - or different algorithms - will provide insight into potential improvements regarding the computational performance of this tool.

Optimizations against observations of annual mean nutrients and oxygen, using different optimization setups could reduce the misfit of the model to some extent; however, they resulted in two different solutions, and the remaining misfit was still $\approx$ 15% of global mean tracer concentrations. The first obstacle might be related to an indeterminacy of the biological parameters, and has been observed in other studies as well; in addition, the misfit function most likely is not informative enough about these parameters. Tests with different misfit functions and components of the misfit may reveal more suitable measures of model skill. The second problem - a rather high remaining misfit - can probably be related to inappropriate, physical or biogeochemical model setup. Therefore, future studies will address the impact of different misfit functions and tunable parameter combinations for constraining the rather uncertain model parameters. It is important to note that observations that provide information about the upper and lower bounds of biological parameters - such as zooplankton grazing and mortality rates - may provide a good guidance for setting up optimization studies, and lower their computational demand.

We expect, however, that, depending on tracer type, distribution, and form of the misfit function (e.g., weighted vs. un-weighted misfit), optimizations may yield quite different solutions for the resulting parameters, and biogeochemical fluxes (see also Evans, 2003). For one and the same model, structure and components of the misfit function, as a measure of model skill, will likely depend on the scientific question we want to address with the model.

Assessment of parameters in biogeochemical ocean models may involve a misfit topography with many local minima, which probably can best be dealt with stochastic and/or evolutionary algorithms. Local minima in the misfit function, particularly when optimizing many ($> 3$) parameters for which there are only few, uncertain observations regarding their potential values, should give rise to a cautious interpretation of global model results. This has also been discussed extensively by Ward et al. (2010), and later by Löptien and Dietze (2015). It remains to be investigated how parameter uncertainties that arise from global optimizations as the one presented here, will map onto model sensitivities when these are run in forward, predictive mode.

## 6 Code availability

The source code of MOPS coupled to TMM, as well as the optimization framework are available as supplement. The most recent TMM source code, forcing, etc. are available under

`https://github.com/samarkhatiwala/tmm.`

## Appendix A: Source code

As research questions may diverge strongly (and therefore, also the different user groups, hardware, biogeochemical models and circulations), we aimed to construct a tool that is as generic and universally applicable as possible, with a high level of portability among different architectures. The model-optimization framework of TMM already comprises new subroutines for data assimilation and cost (misfit) function evaluation, as well as monitor routines to facilitate run-time checks of model state,





and a more generic coupling interface for biogeochemistry. It can thus easily be applied within an optimization framework. While we here focus on the coarse resolution model, we note that the generic structure of the TMM framework allows the user to easily switch between transport matrices, once these are available. Likewise, coupling different biogeochemical models to the framework only requires editing of a (few) interface subroutines. Finally, in principle it should be possible to exchange the

optimization algorithm by any other algorithm, that requires only model misfit as input, and provides a set of parameter files as output.

Reading a parameter file and computation of misfit are two distinct tasks: one may want to only read a set of parameters (which is usually very specific to a particular model), without computing any misfit function. On the other hand, one may only want to compute the misfit, but apply parameters set in the initialization routine. Therefore, these two tasks - although both are

required for optimization - are assigned to different components of the framework: parameter I/O is related more closely to the biogeochemistry itself, and therefor carried out by `external_forcing_mops_biogeochem.c`, and related subroutines. Computation of misfit is a more general task, and therefore invoked by the main driver code, `tmm_main.c`. However, it is also related to the biogeochemical model structure itself, as the mapping of simulated to observed tracers and diagnostics can depend strongly on the biogeochemical model structure. Therefore, files related to misfit computation are also embedded in the

biogeochemical model subroutines. In the following, files that have been added, or are relevant for input of parameter vectors and computation of misfit functions are denoted by an asterisk. An overview of the model structure and layout, with emphasis on those parts that affect computation of biogeochemical fluxes and tracers, optimization and parameter handling is given in Fig. 17.

## A1  MOPS-2.0 biogeochemical subroutines

Most of the biogeochemical subroutines are described in detail the appendix of Kriest and Oschlies (2015). We here only briefly describe the different biogeochemical subroutines, and refer the reader to that website, and to the detailed documentation in the supplementary material that accompanies this manuscript.

As noted in Kriest and Oschlies (2015), the code mainly consists of outer routines, that connect to the TMM and translate to the "3D" circulation, and inner routines that contain the local biogeochemical sources and sinks, and define the biogeochemical

parameters. These routines communicate via common blocks in header files.

`*external_forcing_mops_biogeochem.c` connects biogeochemical subroutines to the TMM, including input and output of files and runtime parameters. It also determines from runtime options whether a parameter file should be read, and its name. Additionally, it assembles the vectors of individual profiles for tracers, diagnostics, and model equivalents for the misfit function into one combined vector to be passed to the main driver code, `tmm_main.c`. It thus provides the basic interface

between a biogeochemical model and the TMM, and calls the following subroutines:

- `mops_biogeochem_copy_data.F`: maps tracer fields back and forth to communicate generically with the basic TMM structure. This new routine facilitates the introduction of new tracers.

- `mops_biogeochem_ini.F`: basic initialization. It calls




- – `*BGC_INI.F`: sets the biogeochemical parameters. Note that in this file we distinguish between parameters that stay fixed, and parameters that depend on parameters which change during optimization, and therefor have to change as well. For example, the stoichiometry for nitrate loss during denitrification depends on the stoichiometric ratio of $O_2$:P for aerobic remineralization (Paulmier et al., 2009). Therefore, if the latter changes, the former will have the recalculated as well. This is carried out by repeated calls to this routine after new parameter vectors have been read.

- – `*mops_biogeochem_set_params.F`: assigns vector of parameters, read by `external_forcing_mops_biogeochem.c` parameters named in `BGC_INI.F`. Each call to this routine is followed by a call to `mops_biogeochem_ini.F` and `BGC_INI.F` (see above).

- – `mops_biogeochem_model.F`: maps tracer fields used in `BGC_MODEL` onto arrays to be passed to `external_forcing_mop` It calls

  - – `*BGC_MODEL.F`: calculation of biogeochemical sources and sinks. It now also assigns state variables to arrays that will be passed to the misfit function.

- – `mops_biogeochem_diagnostics.F`: maps diagnostic output computed in `BGC_MODEL` onto arrays to be passed to `external_forcing_mops_biogeochem.c`

- – `*mops_biogeochem_misfit.F`: maps arrays of simulated tracers for computation of misfit, computed in `BGC_MODEL` onto arrays to be passed to `external_forcing_mops_biogeochem.c`.

Communication between the different modules is carried out mainly via several header files:

- – `mops_biogeochem.h`: introduces subroutines to `external_forcing_mops_biogeochem.c`

- – `*mops_biogeochem_misfit_data.h` communicates parameters and variables related to misfit computation between `external_forcing_mops_biogeochem.c` and main driver code `tmm_main.c`

- – `BGC_PARAMS.h`: communicates biogeochemical parameters between the different model pieces. It also contains the biogeochemical tracer fields (`bgc_tracer`).

- – `BGC_DIAGNOSTICS.h`: passes arrays for diagnostic output. (Omitted from Fig. 17.)

- – `BGC_CONTROL.h`: passes runtime parameters to biogeochemistry. (Omitted from Fig. 17.)

- – `*BGC_MISFIT.h` passes arrays from `BGC_MODEL.F` to `mops_biogeochem_misfit.F`





## A2 Interfacing computation of misfit with the TMM

For a most generic application of the TMM for biogeochemical model optimization we have devised several new subroutines, that facilitate the implementation of any biogeochemical model into the framework. Therefore, the main driver, `tmm_main.c` communicates directly with

5 — `*tmm_misfit.c`: contains initialization of misfit computation (including input of files of observations and weights, as well as reading parameters for misfit from runtime arguments), the misfit function, and its output to either binary or ASCII files. It communicates with the biogeochemical model (`external_forcing_mops_biogeochem.c`) via `*mops_biogeochem_misfit_data.h`. Its subroutines are introduced to the TMM via header file

— `*tmm_misfit.h`

10 One may want to prevent computation of a simulation if during spinup some parameter values or concentrations lead to erroneous (e.g., negative) tracer concentrations. Routine `tmm_monitor.c` may serve as a module to monitor state variables, or other model properties (not used in the current setup presented here).

## A3 Optimization

As noted above, the framework presented here is set up such that a serial script `serial.job` calls the optimization routine (in 15 our case CMAES), which computes a population of size $= \lambda$ of parameter vectors, stored in ASCII files. The same script then calls a parallel script `parallel.job`, which starts $\lambda$ model simulations. During these simulations, the parameter files are read, and a spinup is carried out for each individual setup. The individual model runs then output the misfit function to specified files. When all jobs are finished, script `parallel.job` invokes script `serial.job` again, etc.. Thus, communication between both alternating steps (creation of parameter vectors and computation of resulting misfit function) is carried out 20 by these parameter and misfit files. In addition, file `nIter.txt` keeps track of the progress of optimization, and provides the information which generation is to be computed; it also contains the runtime parameters for the optimizer, CMAES. See information in supplement for more details on how this setup works, and how to specify biogeochemical and optimizer parameters used e.g., in the work presented here.

*Acknowledgements.* This work is a contribution to the DFG-supported project SFB754 and to the research platforms of the DFG cluster 25 of excellence The Future Ocean. We thank Nikolaus Hansen for supporting open access to the CMA-ES code. Parallel supercomputing resources have been provided by the North-German Supercomputing Alliance (HLRN). The authors wish to acknowledge use of the Ferret program of NOAA's Pacific Marine Environmental Laboratory for analysis and graphics in this paper.



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




**Table 1.** Operational constants of the CMA-ES algorithm (cf. Initialization in Algorithm 1).

| Selection and recombination | Step size control | Covariance matrix adaption |
|---|---|---|
| $\lambda = 4 + \lfloor 3 \log n \rfloor$ | $\chi = \sqrt{n}\left(1 - \frac{1}{4n} + \frac{1}{21n^2}\right)$ | $c_c = \frac{4 + \mu_{\text{eff}}/n}{n + 4 + 2\mu_{\text{eff}}/n}$ |
| $\mu = \lfloor \frac{\lambda}{2} \rfloor$ | $c_\sigma = \frac{\mu_{\text{eff}} + 2}{n + \mu_{\text{eff}} + 5}$ | $c_\mu = \min\left(1 - c1,\, 2\frac{\mu_{\text{eff}} + 1/\mu_{\text{eff}} - 2}{(n+2)^2 + \mu_{\text{eff}}}\right)$ |
| $w_i = \frac{\log(\mu + 0.5) - \log(i)}{\sum_{j=1}^{\mu} \log(\mu + 0.5) - \log(j)}$ | $d_\sigma = 1 + c_\sigma$ | $c_1 = \frac{2}{(n+1.3)^2 + \mu_{\text{eff}}}$ |
| $\mu_{\text{eff}} = \frac{\left(\sum_{i=1}^{\mu} w_i\right)^2}{\sum_{i=1}^{\mu} w_i^2} = \frac{1}{\sum_{i=1}^{\mu} w_i^2}$ | | |

**Table 2.** Experimental setup of optimization. "low" and "upp" indicate boundary constraints of the optimizations, respectively. $\lambda$ is the population size of the optimization.

| Name | $R_{-\text{O2:P}}$ | | $I_c$ | | $K_{\text{PHY}}$ | | $\mu_{\text{ZOO}}$ | | $\kappa_{\text{ZOO}}$ | | $b^{\S}$ | |
|---|---|---|---|---|---|---|---|---|---|---|---|---|
| | low | high | low | high | low | high | low | high | low | high | low | high |
| TWIN | 150 | 200 | 4.0 | 48 | 0.0001 | 0.5 | 0.1 | 4.0 | 0.0 | 10.0 | 0.4 | 1.8 |
| OBS-WIDE | 150 | 200 | 4.0 | 48 | 0.0001 | 0.5 | 0.1 | 4.0 | 0.0 | 10.0 | 0.4 | 1.8 |
| OBS-WIDE-20 | 150 | 200 | 4.0 | 48 | 0.0001 | 0.5 | 0.1 | 4.0 | 0.0 | 10.0 | 0.4 | 1.8 |
| OBS-NARR | 150 | 200 | 4.0 | 48 | 0.0001 | 0.5 | 1.0 | 3.0 | 1.6 | 4.8 | 0.4 | 1.8 |
| OBS-NARR-R | 150 | 200 | 4.0 | 48 | 0.0001 | 0.5 | 1.0 | 3.0 | 1.6 | 4.8 | 0.4 | 1.8 |

[§] Note that from $b$ (the optimized parameter) in the model we calculate the rate of vertical increase in sinking speed $a$, always assuming nominal detrital remineralization of $r = 0.05\,\mathrm{d}^{-1}$. The resulting values for $a$ are: 0.058275 (Target (Twin)), 0.125 (Upper) and 0.027778 (Lower).





**Table 3.** Optimization results (evaluations, i.e. number of individuals, $\lambda$, times number of generations, $N$), best model misfit $M_{\mathrm{opt}}$, optimum parameters and their uncertainties. For each model and parameter, the first line gives the optimum parameter, followed by $p_{\min}$ and maximum $p_{\max}$ of all individuals, for which the misfit $M_i$ is $(M_i - M_{opt})/M_{opt} \leq 0.001$. The third line additionally present in brackets the percent of individuals, for which this criterion holds, as well as the range of optimum parameters as percent of the average parameter of the last generation. We also give misfit and parameters of the reference run, against which the twin experiment was optimized.

| Experiment | $\lambda \times N$ | $M_{\mathrm{opt}}$ | $R_{-\mathrm{O2:P}}$ | $I_{\mathrm{c}}$ | $K_{\mathrm{PHY}}$ | $\mu_{\mathrm{ZOO}}$ | $\kappa_{\mathrm{ZOO}}$ | $b$ |
|---|---|---|---|---|---|---|---|---|
| Reference | 1 | 0.529 | 170.0 | 24.0 | 0.0315 | 2.0 | 2.0 | 0.858 |
| TWIN | 2000 | 0.0003 | 170.0 | 24.0 | 0.034 | 2.0 | 3.20 | 0.858 |
| | | | 170 | 24 | 0.033-0.035 | 2.0 | 3.19-3.20 | 0.858 |
| | ($< 1$) | | ($< 1$) | ($< 1$) | (5) | ($< 1$) | ($< 1$) | ($< 1$) |
| OBS-WIDE | 950 | 0.477 | 179.5 | 48.0 | 0.12 | 0.28 | 6.15 | 1.10 |
| | | | 176-182 | 46-49 | 0.09-0.13 | 0.24-0.32 | 4.79-3.37 | 1.08-1.12 |
| | (31) | | (3) | (6) | (32) | (28) | (26) | (4) |
| OBS-WIDE-20 | 3460 | 0.450 | 167.7 | 9.9 | 0.5 | 2.05 | 5.83 | 1.34 |
| | | | 165-171 | 9.6-10.8 | 0.39-0.57 | 2.00-2.52 | 5.37-10.0 | 1.31-1.37 |
| | (64) | | (3) | (12) | (34) | (25) | (79) | (5) |
| OBS-NARR | 1820 | 0.450 | 167.0 | 9.7 | 0.5 | 1.89 | 4.57 | 1.34 |
| | | | 165-170 | 9.0-10.3 | 0.39-0.53 | 1.57-2.02 | 2.95-4.66 | 1.30-1.36 |
| | (39) | | (3) | (14) | (28) | (23) | (37) | (4) |
| OBS-NARR-R | 1400 | 0.450 | 166.7 | 9.6 | 0.5 | 1.76 | 3.82 | 1.34 |
| | | | 165-169 | 8.7-10.1 | 0.44-0.54 | 1.57-1.79 | 2.77-3.90 | 1.31-1.36 |
| | (50) | | (2) | (14) | (19) | (13) | (30) | (3) |





**Table 4.** Global annual fluxes of primary production (PP), grazing (GRAZ), aerobic and anaerobic remineralization of detritus and DOM to nutrients (REM), excretion by zooplankton (EXCR) export production (F120, flux through 120 m), flux through 2030 m (F2030), and benthic burial (BUR), in Pg N y$^{-1}$, for the reference experiment, OBS-WIDE, OBS-WIDE-20 and OBS-NARR (two repeated experiment with different configurations of CMAES). We also show some globally derived, observed estimates. Conversion between different elements was carried out via N:P=16, and C:P=122.

| Experiment | PP | GRAZ | REM | EXCR | F130 | F2030 | BUR |
|---|---|---|---|---|---|---|---|
| Reference | 5.44 | 3.52 | 4.72 | 0.80 | 0.92 | 0.11 | 0.05 |
| OBS-WIDE | 6.20 | 1.24 | 5.94 | 0.25 | 0.81 | 0.06 | 0.02 |
| OBS-WIDE-20 | 7.45 | 4.68 | 6.66 | 1.00 | 1.10 | 0.06 | 0.02 |
| OBS-NARR | 7.52 | 4.74 | 6.65 | 1.10 | 1.10 | 0.06 | 0.02 |
| OBS-NARR-R | 7.58 | 4.77 | 6.65 | 1.19 | 1.10 | 0.06 | 0.02 |
| Observed[§] | 7.68-8.09 | 4.79, 5.71 | - | - | 0.29-1.53 | 0.03-0.07 | 0.02 |

[§] Observed fluxes are from Carr et al. (2006, primary production), Honjo et al. (2008, particle flux), Lutz et al. (2007, particle flux), Dunne et al. (2007, particle flux), Schmoker et al. (2013, primary production, zooplankton grazing excluding/including mesozooplankton grazing) and Wallmann (2010, burial; without shelf and slope region).




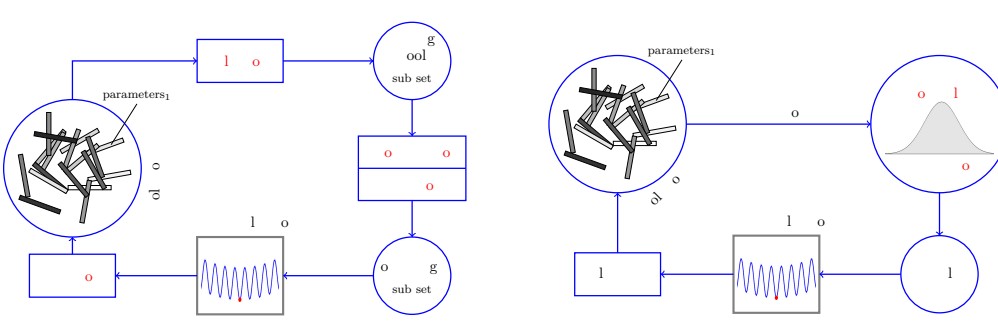

**Figure 1.** A general EA (left) and EDA (right) schematic. EA: A set of candidate solutions (population) is iteratively updated. In each generation, candidate solutions compete to form a mating pool which is realized by a selection operator. Offspring solutions are produced by recombining mates and/or introducing some mutation. Finally, there is a fitness based insertion back into the population. EDA: Candidate solutions of the current iteration's population (and, indirectly, those of former iterations) are used to update an explicit probability distribution such that the likelihood to sample good solutions increases. New samples of the probability distribution replace the current population of candidate solutions

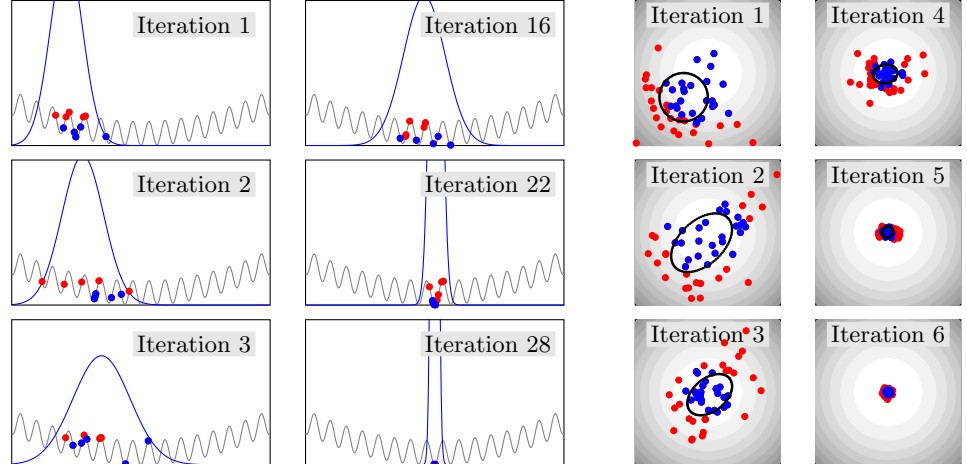

**Figure 2.** Iterations of the CMA-ES applied to test functions. Left: The uni-variate Griewank function (grey curve). In each iteration the normal distribution (blue curve) is sampled 10 times. The samples with their fitness values are shown as dots. The 5 better samples (blue dots) are involved into the normal distribution update for the next iteration. Right: Two-dimensional sphere function. More samples (50) then necessary are used to update the distribution, which is indicated by its standard derivation ellipse (black), here. Distributions tend to elongate into directions of descend (iteration 2). For the convex example function the algorithm converges after few iterations.





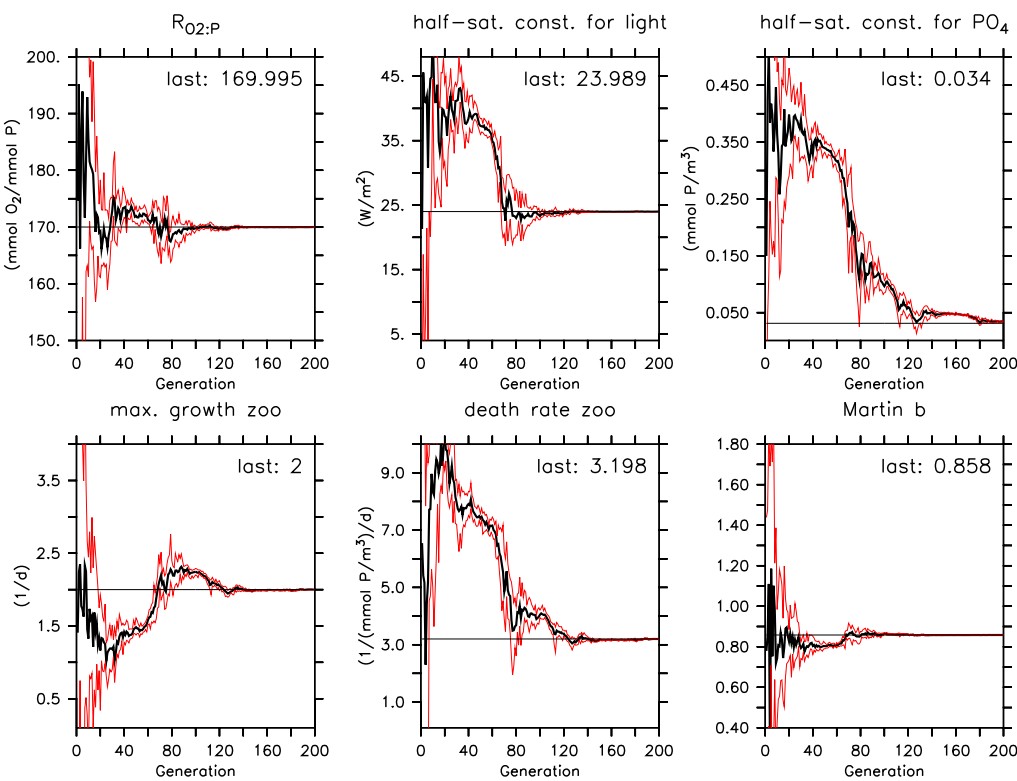

**Figure 3.** Optimization trajectory for six parameters of the twin experiment. Thick black line shows average parameter of all ten individuals of a generation. Red lines indicate their maximum and minimum parameter value. Horizontal black lines indicate the target parameter. Note that we restrict the y-axis to maximum and minimum boundary.



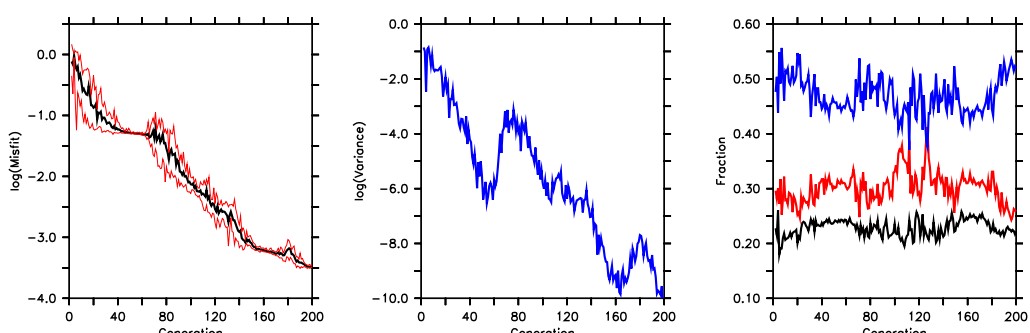

**Figure 4.** Model misfit, its variance, calculated from individuals of each population (both transformed logarithmically) and components of the twin experiment. Left panel: Thick black line shows average misfit of all ten individuals of a generation. Red lines indicate maximum and minimum misfit. Mid panel: Variance of misfit. Right panel: contribution of each component of the misfit Function. Blue: oxygen. Red: nitrate. Black: phosphate.





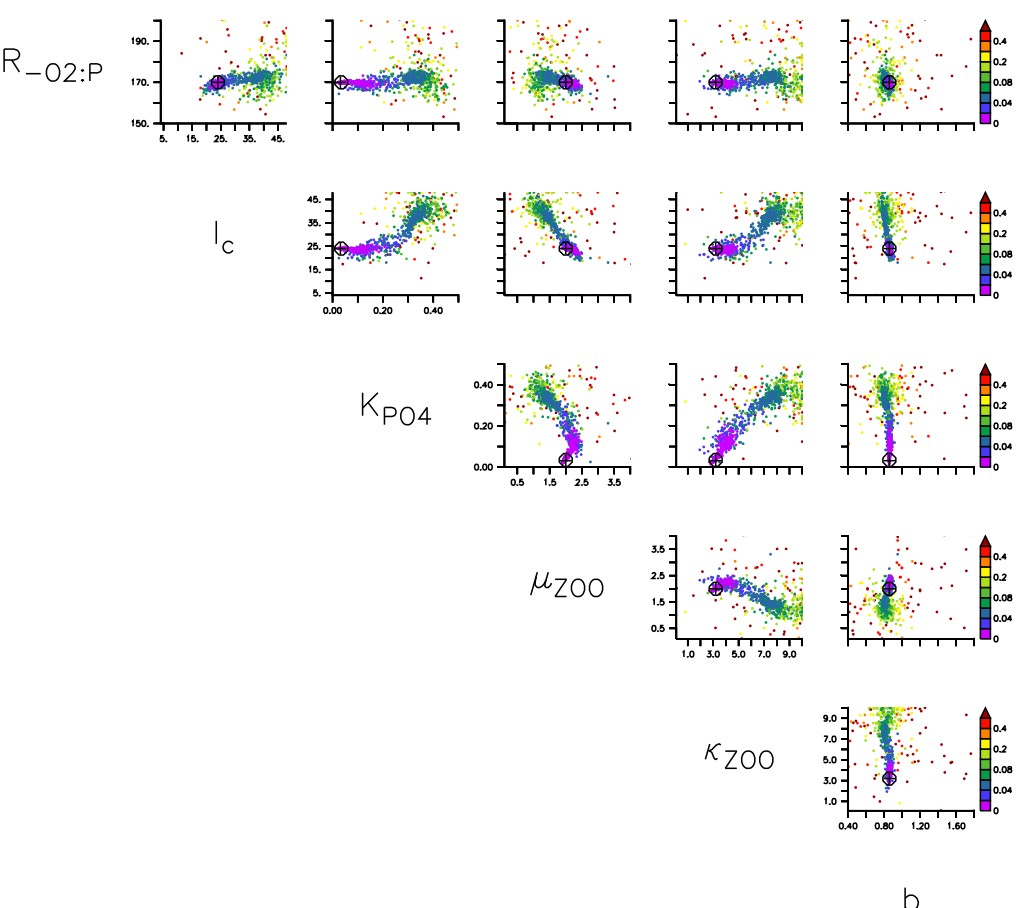

**Figure 5.** Model misfit, plotted for each pair of parameter combinations of the twin experiment. Color indicates misfit (see color bars on the right). A cross indicates the target value, i.e. the value of the reference experiment. A circle indicates the parameter of one individual of the last generation. Note that for better visibility we restrict the parameter range to its boundaries (see Table 2).





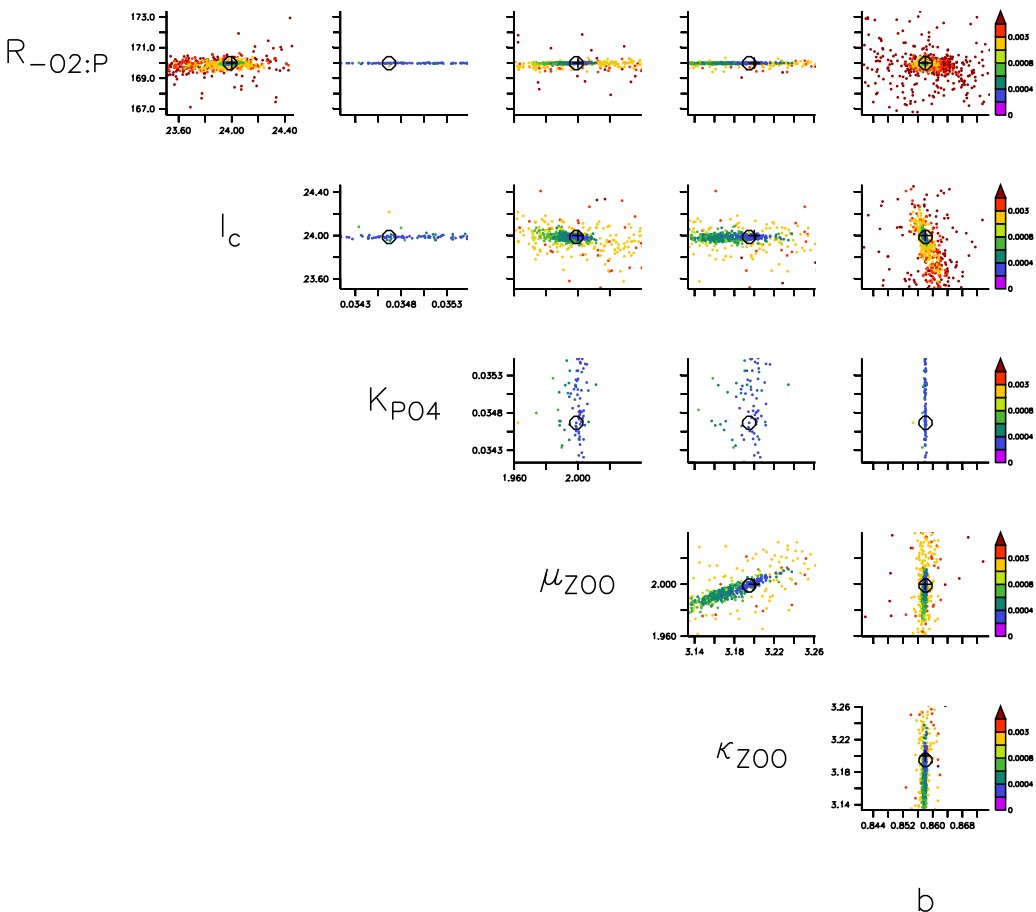

**Figure 6.** As Fig. 5, but only plotted for a region ±2% around the average parameter value of the last generation. Note that the color scale is different than in Fig. 5.

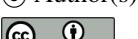



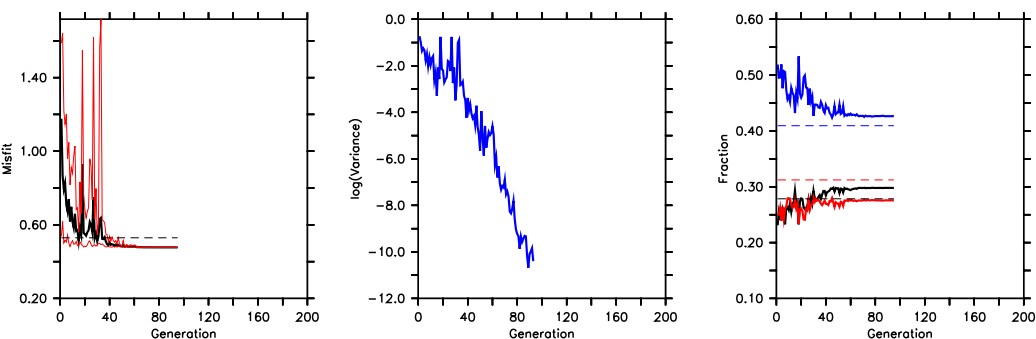

**Figure 7.** As Fig. 4, but for optimization OBS-WIDE. Note that in the left plot, we now show the raw value of the misfit function (not log transformed). The optimization finished at generation 95.

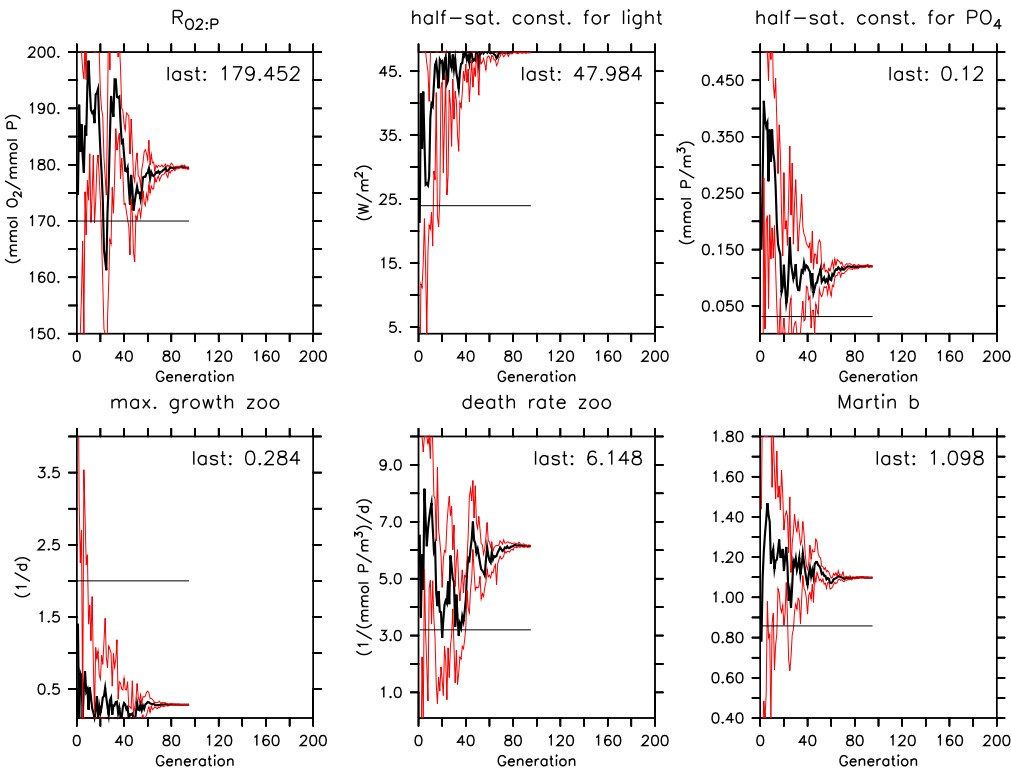

**Figure 8.** As Fig. 3, but for optimization OBS-WIDE. The optimization finished at generation 95.




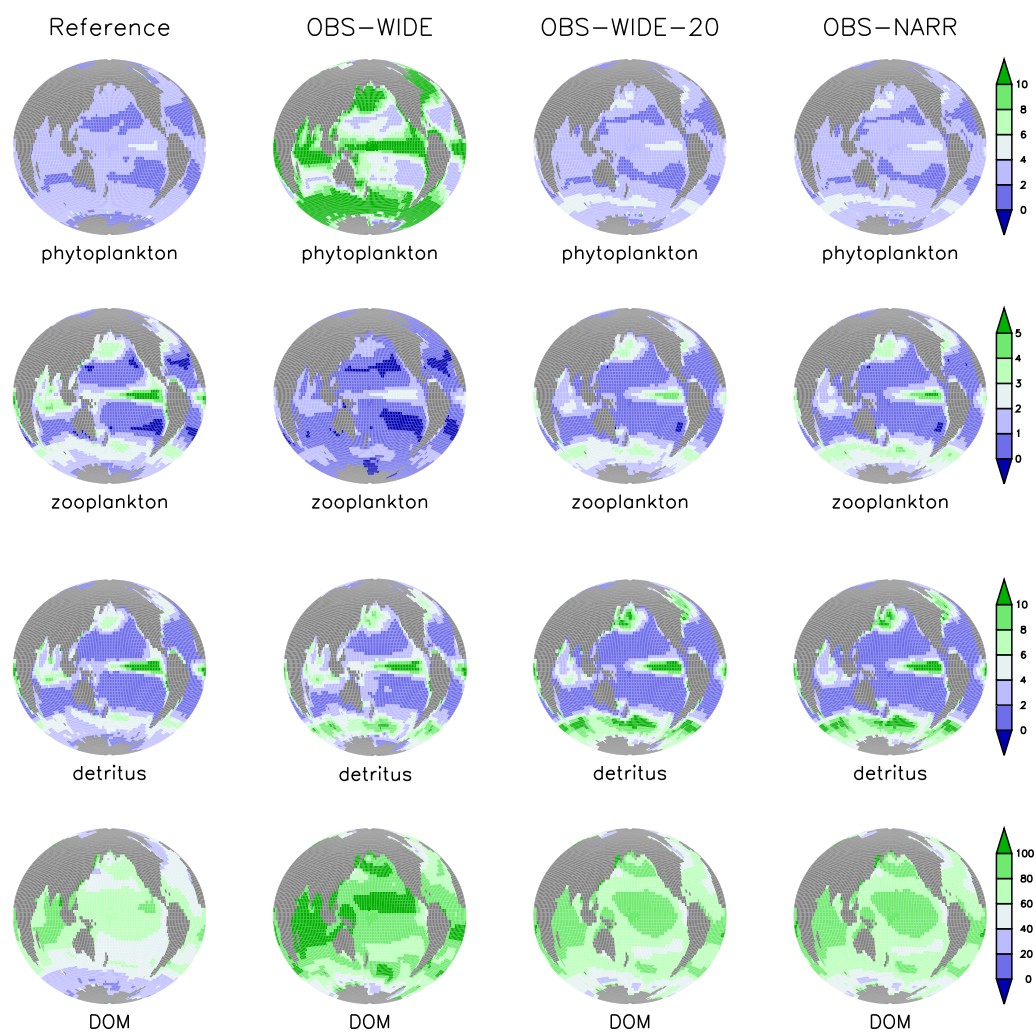

**Figure 9.** Surface (first) layer concentrations (in mmol C m$^{-3}$, converted via a C:P ratio of 122) for phytoplankton, zooplankton, detritus and DOM for the reference run, optimizations OBS-WIDE, OBS-WIDE-20 and OBS-NARR.

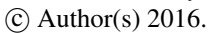



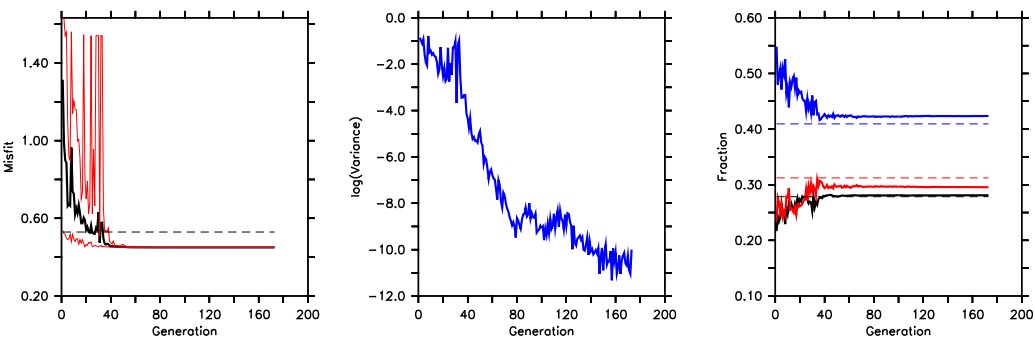

**Figure 10.** As Fig. 7, but for optimization OBS-WIDE-20. The optimization finished at generation 173.

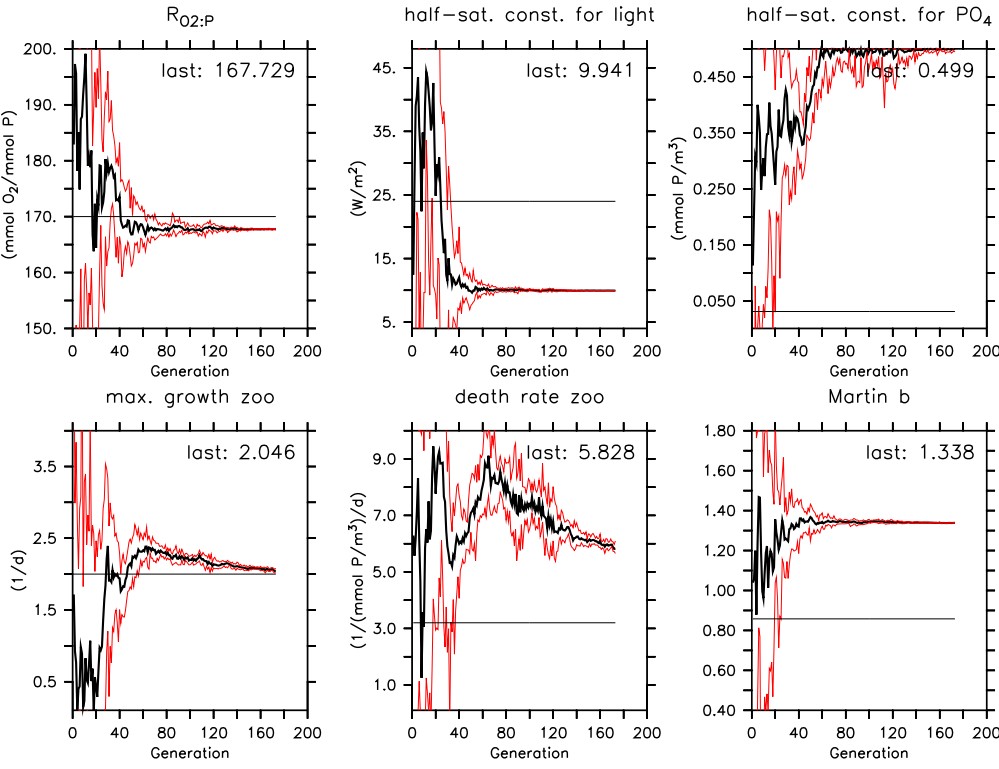

**Figure 11.** As Fig. 8, but for optimization OBS-WIDE-20. The optimization finished at generation 173.





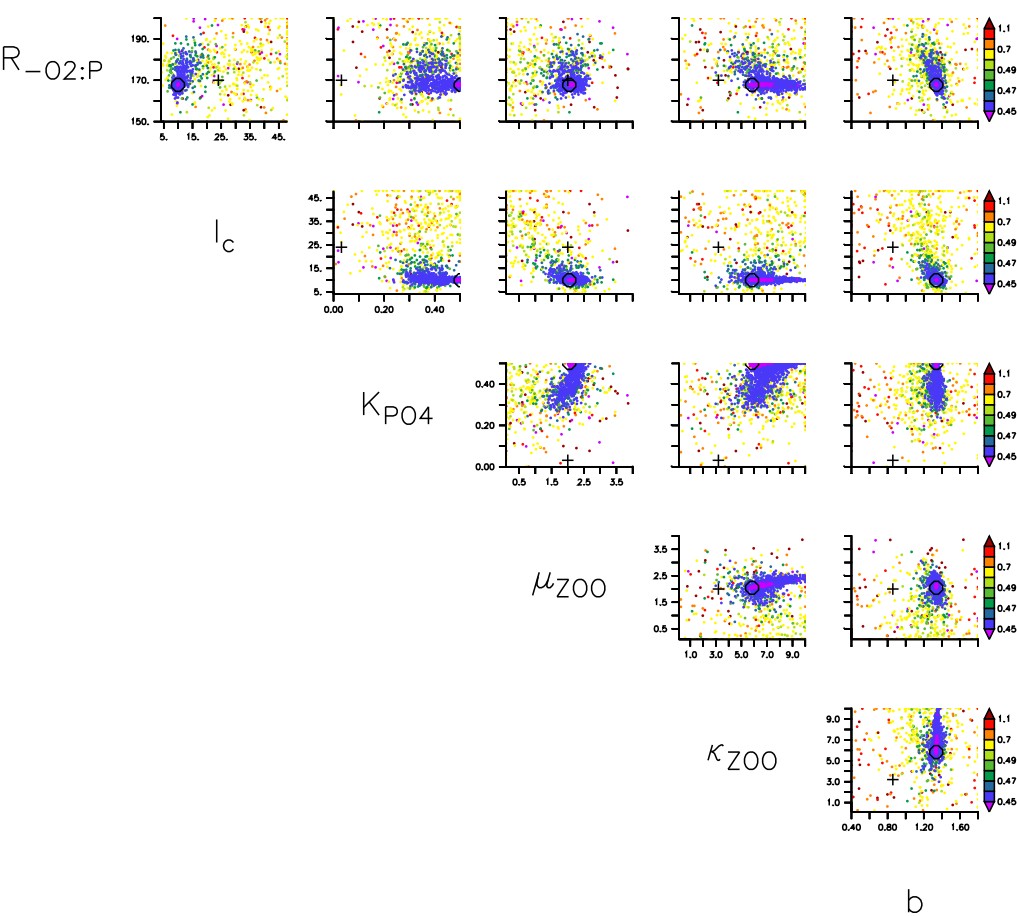

**Figure 12.** As Fig. 5, but for optimization OBS-WIDE-20. Note that the color scale differs from that of Fig. 5.





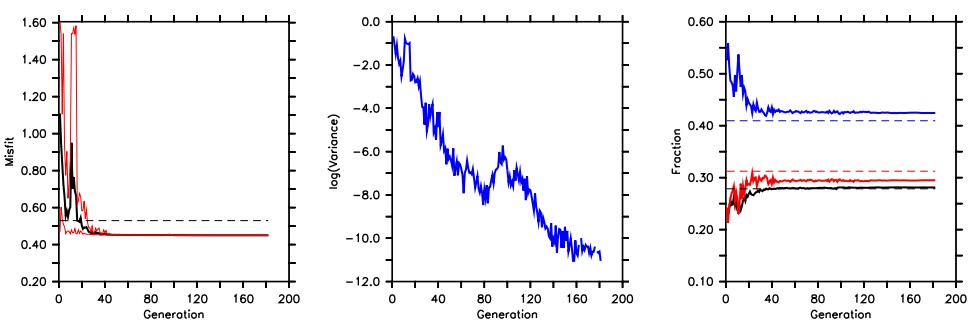

**Figure 13.** As Fig. 10, but for optimization OBS-NARR. The optimization finished at generation 182.





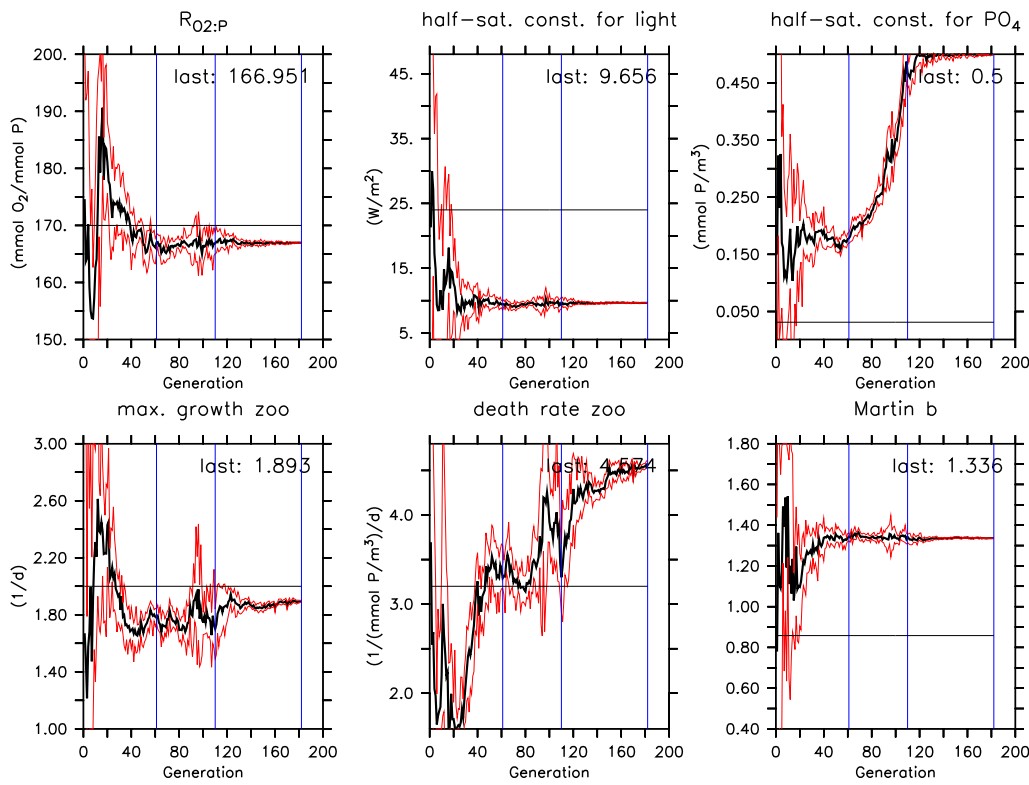

**Figure 14.** As Fig. 11, but for optimization OBS-NARR. The optimization finished at generation 182. Vertical blue lines indicate generation, for which we also present deviations from observation of vertically integrated nutrients and oxygen from in Fig. 16.





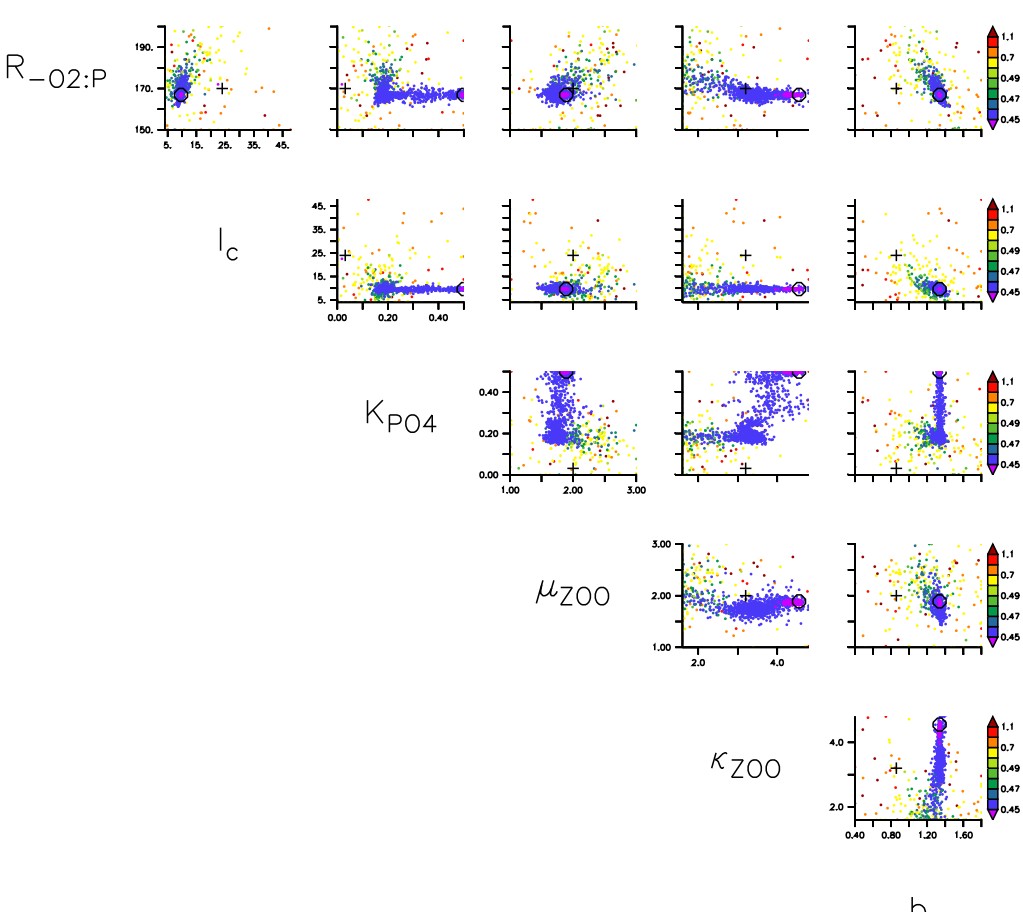

**Figure 15.** As Fig. 12, but for OBS-NARR.




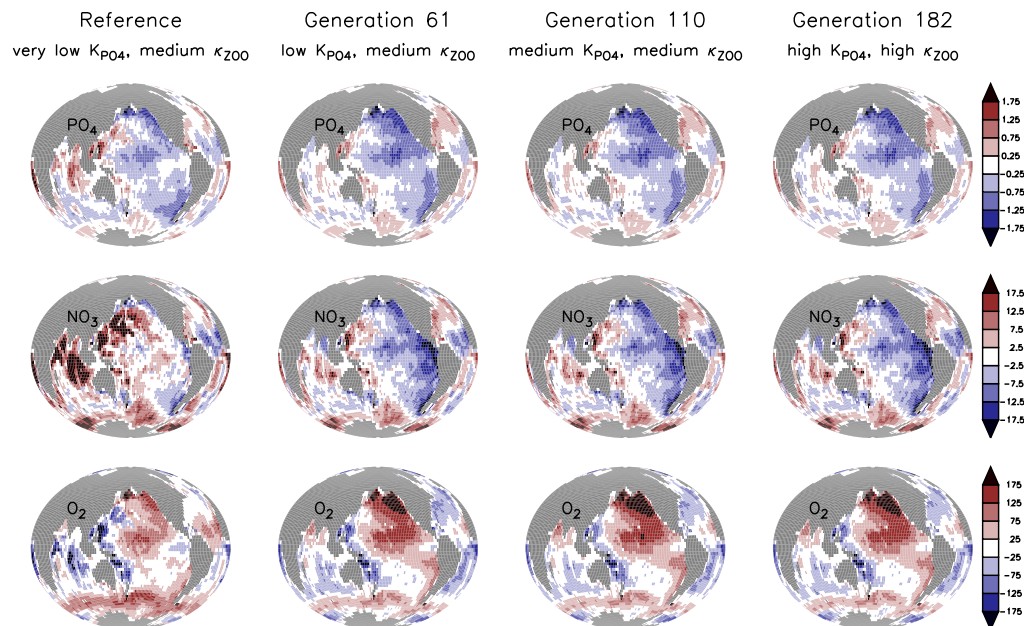

**Figure 16.** Model deviations from observations of vertically integrated phosphate (top), nitrate (middle) and oxygen (bottom) for the reference run, and three generations (61, 110, 182) of OBS-NARR. See blue lines in Fig. 14 for parameter values in this generation. For each generation, we chose the best individual for plotting.





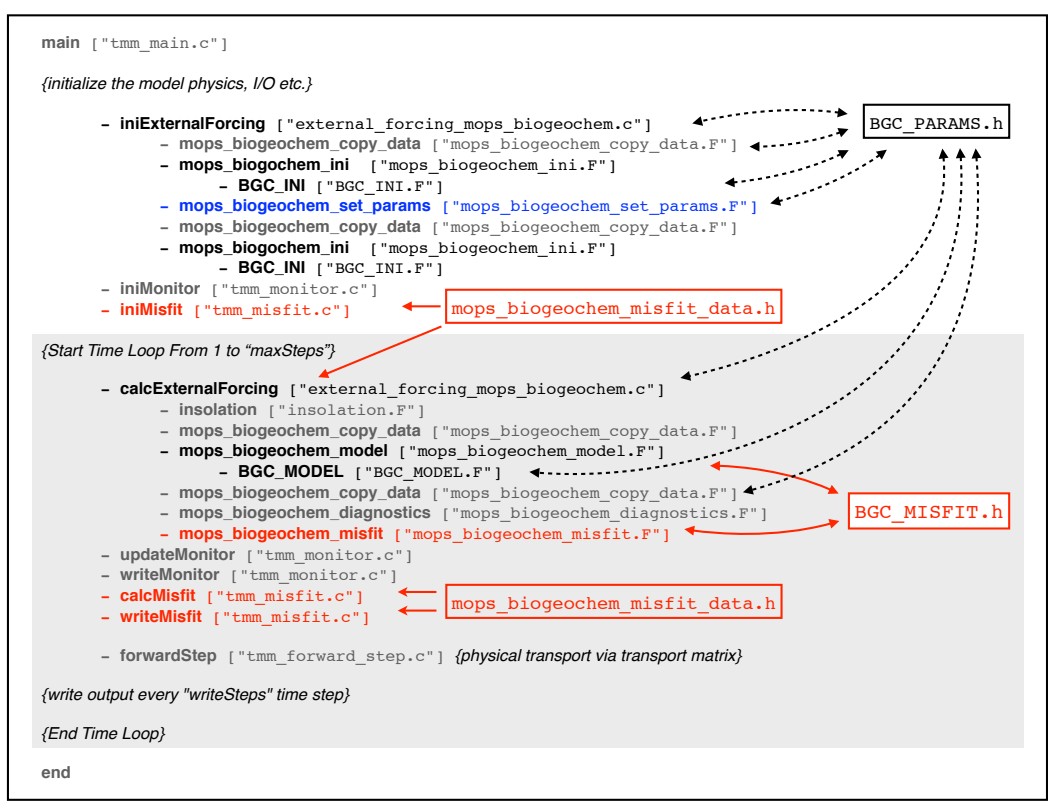

**Figure 17.** Simplified overview over model structure, connection between different subroutines and files, with emphasis on biogeochemical model computation and parameter optimization (subroutines for parameter input highlighted in blue, for misfit function in red).