# Peer review of "Calibrating a global three-dimensional biogeochemical ocean model (MOPS-1.0)"

_Geoscientific Model Development, 2016_

## Referee Comment (RC1) · Anonymous Referee #1 · 4 Oct 2016

These authors demonstrate the use of an evolutionary algorithm to optimize parameters for a marine ecosystem model to observed distributions of nitrate, phosphate and oxygen using offline advection from stored circulation fields. This is a worthwhile experiment, and I congratulate the authors on a (mostly) well written paper.

My most important objection to the present text is that the description of the methodology is somewhat difficult to follow. The authors have tried to present some very mathematical material in accessible terms, and I applaud them for making the effort. But I am not convinced that they have succeeded. This is something that has plagued this field from the beginning. I reviewed some of the very first ocean biogeochemical model parameter-estimation papers, at the very beginning of my reviewing career. I distinctly remember advising authors to try a bit less hard to use 'lay' language, because they weren't really succeeding at making the material accessible or understandable.

[Figure]

Try this: write out the method as if you were writing it for an audience that understands all of the mathematics, using both words and equations and fully integrating the mathematics into the prose, but avoiding jargon as much as possible. Then go through it and look for points where the language could be made more accessible, without leaving the intended meaning vague and without changing the overall structure (much).

I think this exercise will produce a more understandable description of the method, at least understandable by what I would consider the target audience for this paper, i.e., oceanographers like myself who have a basic understanding of, but not a deep familiarity with, the mathematics employed here. To put it bluntly: biological and chemical oceanographers whose comfort level with mathematics does not extend much beyond $Y=a+bX$ are not going to read this paper, or at least they aren't going to read the Methods in any depth. The important thing is to document exactly what was done so that others can reproduce it.

The weighting scheme expressed by $w_i$ could be better explained. From Table 1, Section 2.2.4 and the algorithm schematic (p. 9) we can conclude the following: (a) The w's do not change over the course of the optimization; (b) there is some sort of a priori ranking that allows these weights to be defined us a function of the index i; and (c) the basis for the latter is not explained. Section 2.2.4 states that samples should get more reliable for larger lambda, via a regression-to-the-mean argument. But this logic does not really tell the reader why $w_1 > w_2 > w_3$, when in fact everything presented here is, or could be, for a single value of lambda. As the text is currently written, the rolling a dice (sic) analogy is fatuous. Obviously the sample mean will on average be closer to the population mean for larger samples. But the present text seems to imply that a sample of e.g., n=5 will be more reliable than a sample of the same size if you draw more of them. I don't doubt that the methodology is valid, but the present description is confusing and incomplete.

The captions to Figures 1 and 2 are not very informative. Figure 1 caption does not explain the meanings of the symbols that appear within the shapes or of the shapes

themselves (circles vs rectangles). Figure 2 does not have axis labels. It is difficult to guess what is meant by "fitness values are shown as dots" when these dots fall directly on the function they are sampled from. Is fitness the x or the y axis? I have a hard time envisioning it as either.

Minor points:

(1) The Conclusion is unfocused, meanders among topics, and repeats points already stated in the Discussion. I think it could be cut to about half its current length, if it were clearly focused on what are the key take-home messages of this work.

(2) The sinking model could be better explained, given its significance to the main points of this work. I understand that it is fully explicated in the previous publication cited, but one or two sentences spelling out exactly what are the assumptions and functional relationships used will make it easier on the reader. I would also advise to state what the primary currency of the model is (in section 2.1.2), i.e., are the biological compartments denominated in N or P units (see Figure 9).

(3) The experiment codes in Section 2.6 should be explained, and not just in the sense that the abbreviations are defined. This paragraph should be expanded to include an explanation of what the purposes of the different experiments are, in a conceptual sense. For example, WIDE appears to indicate broad limits on what values the parameters can take in the optimization (vs narrow a priori imposed limits). The reader will eventually figure this out, but it is good practice to clearly state it up front.

(4) Section 3.1 emphasizes the reasons for slow convergence of K_PO4, but glosses over the fact that convergence of the zooplankton growth and death rates is not much faster (Figure 3). It is clear that all 3 of these parameters are quite strongly correlated with each other (Figure 5), so the slow convergence is not very surprising, as the misfit function surface will be more or less flat over a large area of the parameter space. This correlation is also apparent in the subsequent sections (e.g., 13/11-12, 15/3, Table 3).
(5) Figure 6 is not adequately explained. The caption simply states that what is shown here is "a region ±2% around the average parameter value of the last generation" while the text states it shows a region of parameter space "close to the optimum". The most plausible interpretation I can think of here is that "the last generation" represents the one prior to convergence having been declared and the optimization terminated. But this could be spelled out more clearly in the caption.

(6) Figure 16 could use some summary statistics. In some cases it looks as if the optimized parameters are worse than the reference case, but there is a lot of regional variation. It would be good if the global integrals of the misfit function were stated in each case. It would also be a very good idea to include some statement of what defines the 'best' individual.

(7) I don't think Figure 17 is necessary. If this material is really necessary I think it would be better to format it as text, similar to the algorithm schematic on p. 9.

Details:

1/14 change "model's" to "model"

2/6-8 Move Orr ref to the end in same parenthesis as Najjar. Current wording confuses OCMIP1 and OCMIP2, i.e., refers to protocols for OCMIP1 and then cites results from OCMIP2.

2/9 delete "global"

2/19 delete "rather sluggish"

2/26 change "insufficient" to "inappropriate"

2/26-28 "The establishment of an automatic optimization of global biogeochemical ocean models is aimed for in this current study that should enable ..." The development of automatic optimization of global ocean biogeochemical models that is the goal of this study should enable...

[Figure]
3/1 change "environments" to "resolution"

3/16-18 "This efficient "offline" method for ocean passive tracer transport represents the advective and diffusive components of an ocean circulation model in form of transport matrices, that have been extracted prior to the biogeochemical simulations performed here from a physical global circulation model." This efficient "offline" method for ocean passive tracer transport represents advection and mixing in the form of transport matrices that have been calculated from an ocean circulation model simulation prior to the biogeochemical simulations performed here.

3/20-23 I don't think the "see also" or the multiple references to the same paper within the same sentence are necessary.

3/26 MOPS should be defined at first use

4/2-4 "Both aerobic and anaerobic remineralization are parameterized as a saturation curve, using half-saturation constants to regulate the affinity of these processes to either oxidant, as well as the inhibition of denitrification through oxygen." Aerobic and anaerobic remineralization are parameterized as saturation (Monod-type) curves that regulate the rates of these processes using either oxidant, as well as the inhibition of denitrification by oxygen.

4/4 delete "accomplished" or change it to "actual"

4/12 via a parameterization of river runoff? I doubt that this model has explicit river inputs.

4/17 and elsewhere CMAES is sometimes hyphenated, sometimes not

4/29 change "opposite" to "contrast"

4/33 "searchspace" should be "search space" ("eigenvalue", "eigenvector", and "univariate", by contrast, are actual words (see 6/4-9))

5/2-3 "QiEA versions for continuous problems have also been investigated in the literature." Could use a literature reference

5/13 "therefor" (this misspelling appears repeatedly throughout the text)

5/12 "pseudo code" I assume this refers to the algorithm outline, which is useful, but I don't think this term is appropriate here.

5/17-18 "Gaussian bell" I don't think this term is useful or necessary. A Gaussian distribution is sometimes colloquially referred to as "bell curve", but the term is not normally used in the scientific literature. You have defined the distribution as Gaussian, so most subsequent references to "the bell" could just refer to the "the distribution". You might have to finesse the wording in a few places, but I would prefer if this term were not used. "the mean of the bell is attracted towards the good samples" is a good example of the kind of writing I critiqued in my general comments: it tries too hard to be accessible and ends up just being vague.

6/22 I think I understand what sort of vector multiplication is implied here but I'm not sure the terminology is correct (see http://mathworld.wolfram.com/VectorDirectProduct.html). If you multiply x*y' in Matlab for example, it represents a scalar product, which is clearly not what is meant here (see also algorithm outline on p. 9, 3rd to last line of while loop).

6/23 change descend to descent (this misspelling appears several times, in the text and Figure 2 caption)

7/28-29 "the minimum of the penalized fitness function lies within the feasible box" Shouldn't this penalty function be 0 for points inside the boundaries?

11/12 "different random selection of the parameters from the distribution" A different random selection of parameter values; the parameters sampled are fixed.

9/6 Why not state what the "termination criterion" is? (see also "stopping criterion" in algorithm outline above, 11/17, 13/4)

11/31 delete "and large ocean volumes"

12/6 "do not decrease monotonously" monotonically

12/9 delete "obviously"

13/26 Change "a phenomenon that does not occur in the real ocean" to something like "a statistically optimal but physically meaningless solution"?

14/6 "a closer fit to biogeochemical fluxes" based on what? There are no fluxes in the misfit function. Perhaps Table 4 provides support for this assertion but it is not cited.

14/15 "organic tracer concentrations" I think this refers to biological tracers like zooplankton, as opposed to "inorganic" tracers like nitrate (15/12-14). These are sometimes referred to as "abiotic" tracers (15/1). I would suggest just referring to "tracers" generically and "biological" tracers where appropriate, perhaps with "(e.g., phytoplankton)" at first occurrence for clarification. Choose your wording but I strongly recommend that "abiotic" not be used.

14/27 "for some parameters it is quite insensitive to changes" changes in what?

15/15 "not improved on cost of any other tracer" not improved at the cost of any other tracer (see also 16/21)

16/30 change "resembles" to "represents"

17/11 "Another possibility to avoid undesired effects like nearly extinct zooplankton is to bring in further objectives which consider that issues." Another possible way to avoid undesired effects like nearly extinct zooplankton is to introduce further criteria that take account of this issue.

17/12 "the cost function" This term appears out of the blue and is not defined until much later. I don't care if you say cost function or misfit function but be consistent.

17/19 "The topic of multi-objective optimization is intensively regarded" I can't tell what

this means.

17/25 "It remains to be investigated, whether this is related to the lack of temporal solution, or to phosphate not being too tightly related to dissolved or particular organic matter." It remains to be investigated whether this is related to the lack of seasonal data, or to phosphate concentration being weakly dependent on dissolved or particular organic matter concentration.

18/5 change "cure for" to "solution to"

20/12-14 "However, it is also related to the biogeochemical model structure itself, as the mapping of simulated to observed tracers and diagnostics can depend strongly on the biogeochemical model structure." If one is looking for opportunities to shorten the text this would seem to be a good place to start.

20/20 add "in" before "the appendix"

20/21 "refer the reader to that website" Doesn't this refer to a published paper? If it doesn't then we need a lot more detail, because the reader is referred to KO15 for all of the details of the biogeochemical model.

20/28 delete "vectors of"

23/17 Something is wrong here. Why is "reprint of" necessary?

24/12 why is a Discussion paper from 2014 cited? Was the final paper not accepted? (see also Seferian et al)

26/20 Srokosz misspelled

In Table 2 the term used to define the upper boundary differs between the caption (and the footnote) and the table headers.

In Table 3 caption change "brackets" to "parentheses" and delete first comma.

In Table 4 the depth for export is given as 120 m in the caption and 130 in the column

header.

In Figure 2 caption change "then" to "than" and "standard derivation" to "standard deviation"

In Figure 4 caption specify log10 or ln
* * *

---

## Referee Comment (RC2) · M. Butenschön (Referee) · 21 Oct 2016

**1 Referee report for "Calibrating a global three-dimensional biogeochemical ocean model" by Kriest, I. et al., GMD-discussions, 2016**

**1.1 General Comments**

The authors address a highly relevant topic to the modelling of global biogeochemical ocean cycles and marine ecosystem modelling in general, i.e. the optiomisitation of the parameterisation of these models on the base of external constraints in the from of observations of the marine environment. They present a generalised optimisation framework based on state-of-the art components, such as the Transport Matrix Method

to efficiently resolve the advection-diffusion process of ocean tracers and the Estimation of Distribution Algorithms used in the optimisation score offering a highly efficient and generalised tool to the scientific community to address this topic (and others). The authors are to be complemented in their effort in building this system and are discussing the strengths and weaknesses of their method in substantial detail, but before recommanding the work for publication in Geoscientific Model Development, I'd like to see the following points addressed.

**1.1.1 Main points**

- It would be nice to see a test of the optimisation against standard test cases (e.g. Lennart-Jones clusters or similar) in terms of convergence and efficiency with respect to other optimisation methods? The test cases given serve well as an illustration of the procedure, but not as a benchmark. (Maybe some more benchmarks are given in the cited literature that can be referred to?)

**Page 5, line 21 f.:** While the normale distribution is a sensible choice, I wonder if ". . . is considered to provide the best search diversity. . . " reflects the authors oppinion (in which case this should be made clearer by rephrasing) or general consensus (in that case: are there any references?). In addition, a lot of biogeochemical parameters will not be valid for negative values, so a truely symmetric probability density function is unlikely. The Gaussian assumption may still be good enough for the "relevant part" of the parameter space, i.e. the area within the bounding constraints, but maybe this point deserves some consideration.

**Page 5, line 26 f.:** How are the total number of samples and the number of samples to be replaced chosen? How do these choices affect the performance?

**Page 6, line 18 f.:** Again, what is the choice for the weight factors and how does it affect performance?

**Page 7, line 1 f.:** What does the c_mu factor mean for the performance? I'd expect it to slow convergence down. In that case, a discussion of trade-offs between using more samples or a higher c_mu would be interesting.

**Section 2.5:** How is the choice of parameters to be optimised motivated? Especially with respect to surface vs. deep processes and the focus of optimisation on the deep reservoire.

**Page 11, lines 1-5:** Might be worth loosing a couple of words on what kind of configuration/set-up MOPS-RemHigh is. Do I understand this well that the TWIN experiments evualates TMM+MOPS against nutrients fields from a MOPS-RemHigh?

**Page 11, line 21:** Until this point I wasn't sure if the simulations actually did run the full 3000 years for each parameter set candidate or if they used a "collective" spin-up. I'm glad the former is the case, but it might be worth making this point clearer before (section 2.2).

**Page 12, line 14 f.:** Also the global mean of phosphorus in the model is unconstrained, as there's is no constraint on the total amount of biomass in the current optimisation framework as far as I can gather. It is true that there are no global sources or sinks for phoshorus in the model, but that doens't mean that there's no error in the total amount. It just doesn't change during the optimisation (as long as the total phosphorus initial condition isn't included in the optimisation parameters).

**Page 15, line 30:** I thought that the main aspect of the issue of OBS-WIDE was not so much that it was trapped in a local minimum, but that it ended up in an area of the parameter space that yielded unfeasible results on the base of observational constraints not used in the optimisation (e.g. the grazing rates). So in principle, the minumum found may even be global (even if in this case it is not, looking at the misfit values achieved), but the resulting biogeochemical fields and fluxes are unreasonable. To me that is in an important difference, highlighting the fact that the automated optimisation process is not guaranteed to deliver acceptable results, but may still require expert

judgement as additional validation, as long as the observations used in the optimisation process are not sufficient to fully constrain the ecosystem functions modelled. (In fact the misbehaviour might in principle fall into a part of the modelled food-web that is not sufficiently constrained to demonstrate misbehaviour quantitatively, in particular for more complex models).

**Page 16, line 6 f.:** "Increasing the population size . . . ." Reiterating the previous comment, it is not guaranteed that an optimal solution that is judged unfeasible on the base of data or knowledge that is not used in the optimisation process, represents only a local minimum and not a global one. Specifically, there is no reason why this minimum should have higher misfit values than any other minimum within the set of other optimal solutions that deliver reasonable results. So there's actually no guarantee that increasing the population size would help.

**Figure 1:** This figure is hard to understand and needs to be explained better in the caption in order to be useful, e.g. what are the letters? what do the different box shapes (circles vs. squares) represent?

**1.1.2 Minor Comments**

**Page 4, lines 22 ff.:** Might be worth explaining exploitation vs exploration for readers less familiar with the subject of optimasation.

**Page 5, line 4:** Here we use . . .

**Page 5, line 13:** therefore

**Page 5, line 24:** ". . . a some what misuse. . . ", please rephrase to "a misuse to some degree" or similar.

**Page 5, line 30:** towards

**Page 8, line 15:** What is mu_eff?

**Page 8, line 18:** Where does this damping parameter appear from?

**Page 10, line 4:** See the information...

**Page 10, line 5:** ..., e.g.

**Page 11, lines 2-5:** dissolved inorganic oxygen

**Page 11, lines 8-13:** How are the parameter bounds chosen?

**Page 14, line 27:** What is "it"?

**Table 2 heading:** I can't find lambda in the table, so non need to specify it here.

---

## Author Comment (AC1) · 18 Nov 2016

We would like to express our sincere thanks to both referees for their reviews, and their very helpful suggestions. Below are our detailed, point-by-point replies to both referees.

**Reply to referee 1**

We thank Referee 1 for his/her helpful and detailed comments. Below is our detailed reply to the reviewer's suggestions. (Reviewer's comments in blue italics.)

*Try this: write out the method as if you were writing it for an audience that understands all of the mathematics, using both words and equations and fully integrating the mathematics into the prose, but avoiding jargon as much as possible. Then go through it and look for points where the language could be made more accessible, without leaving the intended meaning vague and without changing the overall structure (much).*

We rewrote the section about the optimization methodology. The test now contains more information about the weighting scheme and the relevant information about the backgrounds of the algorithm details. Having removed the "bell" analogy and the "rolling a dice" analogy, it is in a form we would offer to a mathematical audience but also appears generally accessible to us, now. However, as readers who are mainly interested in the application results of the paper might get deterred by the algorithm details, we included a sentence that sections 2.2.2 to 2.3 are for the sake of completeness.

*The weighting scheme expressed by w_i could be better explained. From Table 1, Section 2.2.4 and the algorithm schematic (p. 9) we can conclude the following: (a) The w's do not change over the course of the optimization; (b) there is some sort of a priori ranking that allows these weights to be defined us a function of the index i; and (c) the basis for the latter is not explained. Section 2.2.4 states that samples should get more reliable for larger lambda, via a regression-to-the-mean argument. But this logic does not really tell the reader why w_1>w_2>w_3, when in fact everything presented here is, or could be, for a single value of lambda. As the text is currently written, the rolling a dice (sic) analogy is fatuous. Obviously the sample mean will on average be closer to the population mean for larger samples. But the present text seems to imply that a sample of e.g., n=5 will be more reliable than a sample of the same size if you draw more of them. I don't doubt that the methodology is valid, but the present description is confusing and incomplete.*

We agree with this point. So far, we only mention that the weights are chosen to give samples a rank dependent influence in the distribution update. The extreme cases would be equal weights of 1/mu for all mu better samples and only one weight w1=1 (meaning that only the very best sample is used), respectively. But we skipped the justification for the exact choice of the weights so far. The tutorial of Nikolaus Hansen derives the term "mueff" (see Table 1, first column, last row in our paper) from the weights, with mueff=lambda/4 to indicate an appropriate choice. Indeed, the defined weights approximately satisfy that "target equality". The equality mueff=lambda/4 is actually based on a history of rather complex theoretical considerations (Nikolaus Hansen, pers. comm.). For a candidate fitness function (the "infinite-dimensional sphere function") and equal weights, it has been shown that using the mu=0.27*lambda best samples is "optimal" in the sense that the "expected progress per sample" towards the global optimum is maximized. Hansen considers mueff to be the appropriate equivalent to mu, if rank dependent weights are used and therefore suggests the similar target setting mueff=lambda/4. The optimal weights on the infinite sphere function and hence the optimal value for mueff are also known with non-equal recombination weights. These include non-zero weights for all lambda samples but negative weights for the worse half (hence

doubling the value of mueff), while in practice Hansen does not consider negative weights for updating the mean to be a robust enough choice.

Including literature references we added this brief discussion about the weighting scheme to Section 2.3.1 to which we refer forward from Section 2.2.4 (where the weights are initially introduced). We further added the formulas of the unbiased empirical estimates of the mean vector and the covariance matrix and removed the "rolling a dice analogy".

*The captions to Figures 1 and 2 are not very informative. Figure 1 caption does not explain the meanings of the symbols that appear within the shapes or of the shapes themselves (circles vs rectangles).*

The caption now starts with a sentence about the meaning of the shapes and font colors (red font for operations that involve random decisions). There are also some changes in the figure layout to better indicate the difference between EA and EDA. Since the randomness of the EDA belongs to the sampling of the distribution rather than to the distribution itself, we changed the font colors accordingly. We also removed the function plot symbol from the "fitness evaluation operation" since it does not add much information.

*Figure 2 does not have axis labels. It is difficult to guess what is meant by "fitness values are shown as dots" when these dots fall directly on the function they are sampled from. Is fitness the x or the y axis? I have a hard time envisioning it as either.*

We now added axis labels. For the Griewank function example the scales apply to both, the graph of the Griewank function and the graph of the probability distribution. Therefore, the Griewank function we use is actually a scaled version of the standard Griewank function, so we wrote "Griewank type function" in the caption, now. The phrase "fitness values are shown as dots" is indeed insufficient. In the left example each dot marks a pair (x,f(x)) with the sample and its function value as components. By contrast, the dots in the right example mark the actual (two-dimensional) samples while the counter lines indicate their function/fitness values. We changed the caption accordingly.

*(1) The Conclusion is unfocused, meanders among topics, and repeats points already stated in the Discussion. I think it could be cut to about half its current length, if it were clearly focused on what are the key take-home messages of this work.*

We have shortened and rewritten the conclusion, and hope that it appears more focused now.

*(2) The sinking model could be better explained, given its significance to the main points of this work. I understand that it is fully explicated in the previous publication cited, but one or two sentences spelling out exactly what are the assumptions and functional relationships used will make it easier on the reader. I would also advise to state what the primary currency of the model is (in section 2.1.2), i.e., are the biological compartments denominated in N or P units (see Figure 9).*

We now explain the relation between b and r/a in more detail, and also give the basic currency of the model (phosphorus).

*(3) The experiment codes in Section 2.6 should be explained, and not just in the sense that the abbreviations are defined. This paragraph should be expanded to include an explanation of what the purposes of the different experiments are, in a conceptual sense. For example, WIDE appears to indicate broad limits on what values the parameters can take in the optimization (vs narrow a priori imposed limits). The reader will eventually figure this out, but it is good practice to clearly state it up front.*

We have rewritten this subsection to better explain what the different experiments have been designed for, and how they were set up.

*(4) Section 3.1 emphasizes the reasons for slow convergence of K_PO4, but glosses over the fact that convergence of the zooplankton growth and death rates is not much faster (Figure 3). It is clear that all 3 of these parameters are quite strongly correlated with each other (Figure 5), so the slow convergence is not very surprising, as the misfit function surface will be more or less flat over a large area of the parameter space. This correlation is also apparent in the subsequent sections (e.g., 13/11-12, 15/3, Table 3).*

We agree, that correlation among the parameters may also play a role in the difficulties to constrain K_PO4, and comment on this now in several places (end of sections 3.1, 3.2, 3.2.2; last paragraph of section 4.2).

*(5) Figure 6 is not adequately explained. The caption simply states that what is shown here is "a region ±2% around the average parameter value of the last generation" while the text states it shows a region of parameter space "close to the optimum". The most plausible interpretation I can think of here is that "the last generation" represents the one prior to convergence having been declared and the optimization terminated. But this could be spelled out more clearly in the caption.*

The "best" parameter is the average parameter of the last generation. +- 2% means all parameters that lie within 2\% of that parameter value, regardless of generation and associated misfit. These parameters can have even occurred early in the optimization, and even be associated with a large misfit (that would arise from at least one of the other parameters causing a large misfit). We have changed the caption and text accordingly.

*(6) Figure 16 could use some summary statistics. In some cases it looks as if the optimized parameters are worse than the reference case, but there is a lot of regional variation. It would be good if the global integrals of the misfit function were stated in each case. It would also be a very good idea to include some statement of what defines the 'best' individual.*

We added the values of the misfit functions of the different generations to the figure caption.

*(7) I don't think Figure 17 is necessary. If this material is really necessary I think it would be better to format it as text, similar to the algorithm schematic on p. 9.*

We agree, and moved Fig. 17 to the supplement, for those people interested in the code layout.

Details:

*1/14 change "model's" to "model"* - Corrected.

*2/6-8 Move Orr ref to the end in same parenthesis as Najjar. Current wording confuses OCMIP1 and OCMIP2, i.e., refers to protocols for OCMIP1 and then cites results from OCMIP2.* - Corrected.

*2/9 delete "global"* - Corrected.

*2/19 delete "rather sluggish"* - Corrected.

*2/26 change "insufficient" to "inappropriate"* - Corrected.

*2/26-28 "The establishment of an automatic optimization of global biogeochemical ocean models is aimed for in this current study that should enable ..." The development of automatic optimization of global ocean biogeochemical models that is the goal of this study should enable ...  - Corrected.*

*3/1 change "environments" to "resolution" - Corrected.*

*3/16-18 "This efficient "offline" method for ocean passive tracer transport represents the advective and diffusive components of an ocean circulation model in form of trans- port matrices, that have been extracted prior to the biogeochemical simulations per- formed here from a physical global circulation model." This efficient "offline" method for ocean passive tracer transport represents advection and mixing in the form of transport matrices that have been calculated from an ocean circulation model simulation prior to the biogeochemical simulations performed here. - Corrected.*

*3/20-23 I don't think the "see also" or the multiple references to the same paper within the same sentence are necessary. - Corrected.*

*3/26 MOPS should be defined at first use - Corrected.*

*4/2-4 "Both aerobic and anaerobic remineralization are parameterized as a saturation curve, using half-saturation constants to regulate the affinity of these processes to either oxidant, as well as the inhibition of denitrification through oxygen." Aerobic and anaerobic remineralization are parameterized as saturation (Monod-type) curves that regulate the rates of these processes using either oxidant, as well as the inhibition of denitrification by oxygen. - Corrected.*

*4/4 delete "accomplished" or change it to "actual" - Corrected.*

*4/12 via a parameterization of river runoff? I doubt that this model has explicit river inputs.* - In the model river runoff resupplies buried phosphorus and nitrogen via the volumetric flow rates (Perry et al., 1996) of the world's largest rivers as phosphate and nitrate, as described in Kriest and Oschlies (2013)

*4/17 and elsewhere CMAES is sometimes hyphenated, sometimes not* - We now spell CMA-ES hyphenated throughout the paper.

*4/29 change "opposite" to "contrast" - Corrected.*

*4/33 "searchspace" should be "search space" ("eigenvalue", "eigenvector", and "uni- variate", by contrast, are actual words (see 6/4-9)) - Corrected.*

*5/2-3 "QiEA versions for continuous problems have also been investigated in the literature." Could use a literature reference - Corrected*

*5/13 "therefor" (this misspelling appears repeatedly throughout the text) - Corrected.*

*5/12 "pseudo code" I assume this refers to the algorithm outline, which is useful, but I don't think this term is appropriate here. - We now write "algorithm outline" instead of "pseudo code".*

*5/17-18 "Gaussian bell" I don't think this term is useful or necessary. A Gaussian distribution is sometimes colloquially referred to as "bell curve", but the term is not normally used in the scientific literature. You have defined the distribution as Gaussian, so most subsequent references to "the bell" could just refer to the "the distribution". You might have to finesse the wording in a few places, but I would prefer if this term were not used. "the mean of the bell is attracted towards the good samples" is a good example of the kind of writing I critiqued in my general comments: it tries too hard to be accessible and ends up just being vague.* - We agree, and dispense with "the bell" and refer to "the distribution", instead.

*6/22 I think I understand what sort of vector multiplication is implied here but I'm not sure the terminology is correct (see http://mathworld.wolfram.com/VectorDirectProduct.html). If you multiply x\*y' in Matlab for example, it represents a scalar product, which is clearly not what is meant here (see also algorithm outline on p. 9, 3rd to last line of while loop).* - For column vectors $x=(x\_1,\ldots,x\_n)'$ and $y=(y\_1,\ldots,y\_n)'$ the product $x*y'$ is the matrix $A$ with entries $a\_i\_j = x\_i*y\_j$. We added a sentence on this in the manuscript, after the definiton $C\_emp$.

*6/23 change descend to descent (this misspelling appears several times, in the text and Figure 2 caption).* - Corrected.

*7/28-29 "the minimum of the penalized fitness function lies within the feasible box" Shouldn't this penalty function be 0 for points inside the boundaries?* - Yes, it is. We say so, now. With "penalized fitness function" we mean "the sum of the actual fitness function and the penalty function" . We rewrote this accordingly.

*11/12 "different random selection of the parameters from the distribution" A different random selection of parameter values; the parameters sampled are fixed.* - Corrected.

*9/6 Why not state what the "termination criterion" is? (see also "stopping criterion" in algorithm outline above, 11/17, 13/4)* - This was "hidden" in the last sentence (8/32-33) of Section 2.3.1. As the reader might first resort to the algorithm outline, we placed an additional footnote comment in there.

*11/31 delete "and large ocean volumes"* - Corrected.

*12/6 "do not decrease monotonously" monotonically* - Corrected.

*12/9 delete "obviously"* - Corrected.

*13/26 Change "a phenomenon that does not occur in the real ocean" to something like "a statistically optimal but physically meaningless solution"?* - Done, but we chose "biologically" instead of "physically"

*14/6 "a closer fit to biogeochemical fluxes" based on what? There are no fluxes in the misfit function. Perhaps Table 4 provides support for this assertion but it is not cited.* - We now refer to table 4.

*14/15 "organic tracer concentrations" I think this refers to biological tracers like zooplankton, as opposed to "inorganic" tracers like nitrate (15/12-14). These are sometimes referred to as "abiotic" tracers (15/1). I would suggest just referring to "tracers" generically and "biological" tracers where appropriate, perhaps with "(e.g., phytoplankton)"*

*at first occurrence for clarification. Choose your wording but I strongly recommend that "abiotic" not be used.* - We would prefer to stick with "organic" (plankton, DOM, detritus) and "inorganic" (O2, NO3 and PO4) tracers , and replace  "abiotic" with "inorganic".

*14/27 "for some parameters it is quite insensitive to changes" changes in what?* - Changed to "that it is quite insensitive to changes in some parameters"

*15/15 "not improved on cost of any other tracer" not improved at the cost of any other tracer (see also 16/21)* - Corrected.

*16/30 change "resembles" to "represents"* - Corrected.

*17/11 "Another possibility to avoid undesired effects like nearly extinct zooplankton is to bring in further objectives which consider that issues." Another possible way to avoid undesired effects like nearly extinct zooplankton is to introduce further criteria that take account of this issue.* - Corrected.

*17/12 "the cost function" This term appears out of the blue and is not defined until much later. I don't care if you say cost function or misfit function but be consistent.* - Changed ``cost function'' to ``misfit function'' throughout the text.

*17/19 "The topic of multi-objective optimization is intensively regarded" I can't tell what this means.* - We changed it to "Multi-objective optimization is essentially addressed with ..."

*17/25 "It remains to be investigated, whether this is related to the lack of temporal solution, or to phosphate not being too tightly related to dissolved or particular organic matter." It remains to be investigated whether this is related to the lack of seasonal data, or to phosphate concentration being weakly dependent on dissolved or particular organic matter concentration.* - Corrected.

*18/5 change "cure for" to "solution to"* - Corrected.

*20/12-14 "However, it is also related to the biogeochemical model structure itself, as the mapping of simulated to observed tracers and diagnostics can depend strongly on the biogeochemical model structure." If one is looking for opportunities to shorten the text this would seem to be a good place to start.* We have restructured the appendix about model description.

*20/20 add "in" before "the appendix"* - Corrected.

*20/21 "refer the reader to that website" Doesn't this refer to a published paper? If it doesn't then we need a lot more detail, because the reader is referred to KO15 for all of the details of the biogeochemical model.* - Indeed, a reference to that paper and its supplement is sufficient. We have changed the text accordingly.

*20/28 delete "vectors of"* - Corrected.

*23/17 Something is wrong here. Why is "reprint of" necessary?* - Corrected.

*24/12 why is a Discussion paper from 2014 cited? Was the final paper not accepted? (see also Seferian et al)* - This reference somehow survived from very early version of this paper. Changed to reference to final paper.

*26/20 Srokosz misspelled - Corrected.*

*In Table 2 the term used to define the upper boundary differs between the caption (and the footnote) and the table headers. - Corrected.*

*In Table 3 caption change "brackets" to "parentheses" and delete first comma. - Corrected.*

*In Table 4 the depth for export is given as 120 m in the caption and 130 in the column header - Corrected.*

*In Figure 2 caption change "then" to "than" and "standard derivation" to "standard deviation" - Corrected.*

*In Figure 4 caption specify log10 or ln  - Corrected.*

[revised manuscript text omitted]

---

## Author Comment (AC2) · 18 Nov 2016

We would like to express our sincere thanks to both referees for their reviews, and their very helpful suggestions. Below are our detailed, point-by-point replies to both referees.

**Reply to referee 2**

We thank Momme Butenschön for his encouraging and constructive comments. Below is our detailed reply to his suggestions. (Reviewer's comments in blue italics.)

*1.1.1 Main points*

*It would be nice to see a test of the optimisation against standard test cases (e.g. Lennart-Jones clusters or similar) in terms of convergence and efficiency with respect to other optimisation methods? The test cases given serve well as an illustration of the procedure, but not as a benchmark. (Maybe some more benchmarks are given in the cited literature that can be referred to?)*

We now also refer to the report (Hansen et al. 2009a) that describes the testbed of 24 benchmark functions which have been considered in the comparison study of 31 algorithms in Hansen et al., 2010. We also mention its message concerning CMA-ES but would prefer to go without our own additional benchmark function studies, here.

*Page 5, line 21 f.: While the normale distribution is a sensible choice, I wonder if ". . . is considered to provide the best search diversity..." reflects the authors oppinion (in which case this should be made clearer by rephrasing) or general consensus (in that case: are there any references?). In addition, a lot of biogeochemical parameters will not be valid for negative values, so a truely symmetric probability density function is unlikely. The Gaussian assumption may still be good enough for the "relevant part" of the parameter space, i.e. the area within the bounding constraints, but maybe this point deserves some consideration.*

Actually, the normal distribution "maximizes" an index of diversity, the so called entropy. Including references, we rephrased the "diversity sentence". We also mention the "invalid samples issue" at the and of the section about sampling (2.2.3), referring forward to the boundary handling procedure (section 2.2.7) of the algorithm, now.

*Page 5, line 26 f.: How are the total number of samples and the number of samples to be replaced chosen? How do these choices affect the performance?*

We add (in parentheses) that the number deviates from the suggested CMA-ES default setting referring to the algorithm outline section. As drawing more samples increases the exploration capability of the algorithm (the chance that it does not miss good regions of the search space) but also the computational costs, we state so now at this place.

*Page 6, line 18 f.: Again, what is the choice for the weight factors and how does it affect performance?*

This is indeed a fair question since, so far, we only mention that the weights are chosen to give samples a rank dependent influence in the distribution update. The extreme cases would be equal weights of 1/mu for all mu better samples and one weight w1=1 only (meaning that only the very best sample is used), respectively. But what is the background for the exact choice of weights in CMA-ES? The tutorial of Nikolaus Hansen derives the term "mueff" (see Table 1, first column, last row in our paper) from the weights and states that mueff=lambda/4 is considered to indicate an appropriate choice. Indeed, the defined weights approximately satisfy that "target equality".

The equality mueff=lambda/4 is actually based on a history of rather complex theoretical considerations (Nikolaus Hansen, pers. comm.). For a candidate fitness function (the "infinite-dimensional sphere function") and equal weights, it has been shown that using the mu=0.27*lambda best samples is "optimal" in the sense that the "expected progress per individual" towards the global optimum is maximized. Hansen considers mueff to be the appropriate equivalent to mu, if rank dependent weights are used and therefore suggests the similar target setting mueff=lambda/4. The optimal weights on the infinite sphere function and hence the optimal value for mueff are also known for non-equal recombination weights. These include non-zero weights for all lambda samples but negative weights for the worse half (hence doubling the value of mueff), while in practice Hansen does not consider negative weights for updating the mean to be a robust enough choice.

A discussion of these aspects is included into the paper now, referring to corresponding publications.

*Page 7, line 1 f.: What does the c_mu factor mean for the performance? I'd expect it to slow convergence down. In that case, a discussion of trade-offs between using more samples or a higher c_mu would be interesting.*

Yes, the intention is that the information of earlier samples fades out slowly such that the current distribution estimate cumulates information of several iterations samples in order to be more reliable with a small number of samples per iteration. The smaller the factor c_mu is the more former samples contribute to the current distribution estimate, slowing down learning but being more reliable with less samples per iteration. E.g., for our parameter optimization experiments (n=6 and lambda=10) and the given c_mu setting (Table 1), the samples of 23 iterations contribute 63% of the over all information n C, if only Eq. (1) is used to update C. We therefore add the two sentences after introducing the backward time horizon of floor( 1 / c_mu ).

*Section 2.5: How is the choice of parameters to be optimised motivated? Especially with respect to surface vs. deep processes and the focus of optimisation on the deep reservoire.*

We aimed to consider six parameters for optimization, that encompass a large range of time  scales, as well as different trophic levels, vertical domains and dependencies between internal (interactions between compartments) and external (dependence on light) factors . We further aimed to avoid simultaneous optimization of parameters that are obviously related to each other, such as maximum growth rates and half-saturation constants, or sinking speed and remineralization rate. We have added a few sentences to clarify our choice of parameters.

*Page 11, lines 1-5: Might be worth loosing a couple of words on what kind of configuration/ set-up MOPS-RemHigh is. Do I understand this well that the TWIN ex- periments evualates TMM+MOPS against nutrients fields from a MOPS-RemHigh?*

RemHigh refers to a high affinity of oxic and suboxic remineraliztion to oxidants. We have added this to the text.

*Page 11, line 21: Until this point I wasn't sure if the simulations actually did run the full 3000 years for each parameter set candidate or if they used a "collective" spin-up. I'm glad the former is the case, but it might be worth making this point clearer before (section 2.2).*

This is mentioned in section 2.1.1, but we now also mention it in the abstract.

*Page 12, line 14 f.: Also the global mean of phosphorus in the model is unconstrained, as there's is no constraint on the total amount of biomass in the current optimisation*

*framework as far as I can gather. It is true that there are no global sources or sinks for phoshorus in the model, but that doens't mean that there's no error in the total amount. It just doesn't change during the optimisation (as long as the total phosphorus initial condition isn't included in the optimisation parameters).*

Because the global model is mass-converving with respect to sources and sinks of phosphorus (any gain or loss in biomass=organic P is accounted for in the loss/gain of phosphate; buried P will be resupplied by river runoff in the following year), and each simulation starts with the same initial condition, the total global phosphorus mass is constrained.  In contrast, there is an unlimited source/sink of oxygen and nitrogen in the atmosphere, which may exchange with the ocean via air-sea gas exchange or nitrogen fixation, respecively.

*Page 15, line 30: I thought that the main aspect of the issue of OBS-WIDE was not so much that it was trapped in a local minimum, but that it ended up in an area of the parameter space that yielded unfeasible results on the base of observational constraints not used in the optimisation (e.g. the grazing rates). So in principle, the minumum found may even be global (even if in this case it is not, looking at the misfit values achieved), but the resulting biogeochemical fields and fluxes are unreasonable.*

Indeed, the minimum misfit of OBS-WIDE was  about 6% larger than any misfit of the other optimizations against observations (see table 3); thus we concluded that this is a local minimum.

*To me that is in an important difference, highlighting the fact that the automated optimisation process is not guaranteed to deliver acceptable results, but may still require expert judgement as additional validation, as long as the observations used in the optimisation process are not sufficient to fully constrain the ecosystem functions modelled. (In fact the misbehaviour might in principle fall into a part of the modelled food-web that is not sufficiently constrained to demonstrate misbehaviour quantitatively, in particular for more complex models).*

We agree, and this is exactly what we meant to say. Further, a local (or global) minimum always relates to a particular misfit function; the occurrence of local minima with regard to certain observations may point towards an unconstrained parameter. This is one of the reasons why we aim to extend the data sets (if we want to constrain zooplankton) or apply tools such as multi-objective optimization.

*Page 16, line 6 f.: "Increasing the population size . . . ." Resiterating the previous comment, it is not guaranteed that an optimal solution that is judged unfeasible on the base of data or knowledge that is not used in the optimisation process, represents only a local minimum and not a global one. Specifically, there is no reason why this minimum should have higher misfit values than any other minimum within the set of other optimal solutions that deliver reasonable results. So there's actually no guarantee that increasing the population size would help.*

See above: the misfit of OBS-WIDE is indeed relatively large, and most of the parameters differ from those experiments with narrower boundaries, or a larger population size. We agree, that this does not guarantee that the latter optimizations have found a global minimum; but it is more likely, increasing our confidence in either of these setups (larger population size or narrower boundaries).

We would like to note that, when plotting the PDF of the "best" parameters (i.e., all individuals with a misfit not higher than 1% of the minimum misfit) we find bimodal distributions of kappa_zoo (the quadratic mortality rate). One of the modes vanishes if we decrease the deviation from minimum misfit further (i.e., account for all individuals with misfits not higher than 1.001 times minimum misfit). This raises several questions about

the parameter identifiability for zooplankton parameters, and is discussed in detail in Schartau et al. (2016; section 9 and Fig. 8).

*Figure 1: This figure is hard to understand and needs to be explained better in the caption in order to be useful, e.g. what are the letters? what do the different box shapes (circles vs. squares) represent?*
The caption now starts with a sentence about the meaning of the shapes and font colors (red font for operations that involve random decisions). There are also some changes in the figure layout to better indicate the difference between EA and EDA. Since the randomness of the EDA belongs to the sampling of the distribution rather than to the distribution itself, we changed the font colors accordingly. We also removed the function plot symbol from the "fitness evaluation operation" since it does not add much information.

*1.1.2 Minor Comments*

*Page 4, lines 22 ff.: Might be worth explaining exploitation vs exploration for readers less familiar with the subject of optimasation.* We added explanations in parentheses.

*Page 5, line 4: Here we use ...* Corrected.

*Page 5, line 13: therefore* Corrected.

*Page 5, line 24: ". . . a some what misuse. . . ", please rephrase to "a misuse to some degree" or similar.* Corrected.

*Page 5, line 30: towards* Corrected.

*Page 8, line 15: What is mu_eff?* We refer to Table 1 once more. The meaning of mu_eff as a quality measure for the chosen weights (see above) is now shortly introduced with corresponding references.

*Page 8, line 18: Where does this damping parameter appear from?* The factor was also defined in Table1. As it is simply "1+c_sigma" for the selected weights we now prefer to substitute that parameter by "1+c_sigma" in the corresponding place in the algorithm outline.

*Page 10, line 4: See the information. . .* Corrected.

*Page 10, line 5: ..., e.g.* Corrected.

*Page 11, lines 2-5: dissolved inorganic oxygen* Changed this, but added in parentheses: "(herafter termed as and compared to nitrate)"

*Page 11, lines 8-13: How are the parameter bounds chosen?* We have now added two paragraphs on the choice on boundaries in subsection 2.5.

*Page 14, line 27: What is "it"?* Replaced by "the misfit"

*Table 2 heading: I can't find lambda in the table, so non need to specify it here.* Corrected.

**Additional Note**

We changed the first sentence in Section 2.3.1 from "The CMA-ES approach described in Subsection 2.2.1 ..." appropriately to "The CMA-ES approach described in Subsecion 2.2 ...". Therefore the headline of Section 2.2 changed from "Optimization" to "The optimization algorithm CMA-ES" and the headline of Section 2.2.1 from "The optimization algorithm CMA-ES" to "Population-based search heuristics"

[revised manuscript text omitted]